# Retrieval of tropospheric aerosol, NO$_2$ and HCHO vertical profiles from MAX-DOAS observations over Thessaloniki, Greece: Intercomparison and validation of two inversion algorithms

Dimitris Karagkiozidis[1], Martina Michaela Friedrich[2], Steffen Beirle[3], Alkiviadis Bais[1], François Hendrick[2], Kalliopi Artemis Voudouri[1], Ilias Fountoulakis[4,1], Angelos Karanikolas[5,6], Paraskevi Tzoumaka[7], Michel Van Roozendael[2], Dimitris Balis[1], and Thomas Wagner[3]

[1]Laboratory of Atmospheric Physics, Aristotle University of Thessaloniki, Thessaloniki, Greece
[2]Royal Belgian Institute for Space Aeronomy (BIRA-IASB), Brussels, Belgium
[3]Max Planck Institute for Chemistry, Mainz, Germany
[4]Institute for Astronomy, Astrophysics, Space Applications and Remote Sensing, National Observatory of Athens (IAASARS/NOA), 15236 Athens, Greece
[5]Physikalisch-Meteorologisches Observatorium Davos, World Radiation Center (PMOD/WRC), Dorfstrasse 33, 7260 Davos Dorf, Switzerland
[6]ETH Zurich-Institute for Particle Physics and Astrophysics, Hönggerberg campus, Stefano-Franscini-Platz 5, 8093 Zurich, Switzerland
[7]Municipality of Thessaloniki, Department of Environment, Thessaloniki, Greece

**Correspondence:** Dimitris Karagkiozidis (dkaragki@auth.gr)

**Abstract.** In this study we focus on the retrieval of aerosol and trace gas vertical profiles from Multi-Axis Differential Optical Absorption Spectroscopy (MAX-DOAS) observations for the first time over Thessaloniki, Greece. We use two independent inversion algorithms for the profile retrievals: The Mexican MAX-DOAS Fit (MMF) and the Mainz Profile Algorithm (MAPA). The former is based on the Optimal Estimation Method (OEM), while the latter follows a parameterization approach. We

evaluate the performance of MMF and MAPA and we validate their retrieved products with ancillary data measured by other co-located reference instruments. The trace gas differential slant column densities (dSCDs), simulated by the forward models, are in good agreement, except for HCHO, where larger scatter is observed due to the increased spectral noise of the measurements in the UV. We find an excellent agreement between the tropospheric column densities of NO$_2$ retrieved by MMF and MAPA (Slope $= 1.009$, Pearson's correlation coefficient $R = 0.982$) and a good correlation for the case of HCHO ($R = 0.927$). For

aerosols, we find better agreement for the aerosol optical depths (AODs) in the visible (i.e., at 477 nm), compared to the UV (at 360 nm) and we show that the agreement strongly depends on the O$_4$ scaling factor that is used in the analysis. The agreement for NO$_2$ and HCHO near-surface concentrations is similar to the comparison of the integrated columns with slightly decreased correlation coefficients. The seasonal mean vertical profiles that are retrieved by MMF and MAPA are intercompared and the seasonal variation of all species along with possible sources is discussed. The AODs retrieved by the MAX-DOAS are

validated by comparing them with AOD values measured by a CIMEL sun-photometer and a Brewer spectrophotometer. Four different flagging schemes were applied to the data in order to evaluate their performance. Qualitatively, a generally good agreement is observed for both wavelengths, but we find a systematic bias from the CIMEL and Brewer measurements, due

to the limited sensitivity of the MAX-DOAS in retrieving information at higher altitudes, especially in the UV. An in-depth validation of the aerosol vertical profiles retrieved by the MAX-DOAS is not possible since only in very few cases the true
aerosol profile is known during the period of study. However, we examine four cases, where the MAX-DOAS provided a generally good estimation of the shape of the profiles retrieved by a co-located multi-wavelength lidar system. The $NO_2$ near-surface concentrations are validated against in situ observations and the comparison of both MMF and MAPA revealed good agreement with correlation coefficients of $R = 0.78$ and $R = 0.73$, respectively. Finally, the effect of the $O_4$ scaling factor is investigated by intercomparing the integrated columns retrieved by the two algorithms and also by comparing the AODs
derived by MAPA for different values of the scaling factor with AODs measured by the CIMEL and the Brewer.

## 1   Introduction

The planetary boundary layer (PBL), also called atmospheric boundary layer, is defined as the lowermost layer of the tropo-sphere that is directly influenced by the terrestrial surface. The PBL height, at mid-latitudes, expands typically up to $1 - 2$ km during daytime (von Engeln and Teixeira, 2013) and its composition has a strong impact on weather, climate and air quality.
The increasing interest of understanding the PBL's structure and dynamics is apparent in various research fields, from air pol-lution analysis to weather prediction and thus, continuous ground-based monitoring of both chemical composition and aerosol content of the PBL with high temporal resolution is of great importance.

Thessaloniki is a Mediterranean city and it is the second largest city of Greece, located in the northern part of the country. Thessaloniki hosts approximately 10% of the country's total population with more than 1 million inhabitants (Resident Popu-
lation Census, 2011) and with approximately 20% of the country's industrial activity, it is considered one of the largest urban agglomerations in the Balkans (Moussiopoulos et al., 2009). The air pollution sources in Thessaloniki are mainly industrial activities in the western part of the city, road transport and domestic heating during the cold period of the year, while the air quality of the city is affected by local topographic and meteorological characteristics (Poupkou et al., 2011; Kassomenos et al., 2011). Nitrogen oxides ($NO_x = NO + NO_2$), formaldehyde (HCHO) and aerosols are considered major atmospheric pollutants
contained in the PBL of the city.

Nitrogen dioxide ($NO_2$) and HCHO are two important trace gas species of the atmosphere that play a critical role in tropo-spheric photochemistry (Seinfeld et al., 1998), participating in the formation of tropospheric ozone ($O_3$), while aerosols can have a strong influence on air quality and climate through effects on radiation (IPCC, 2007). Both $NO_2$ and HCHO are toxic to humans in high concentrations and can lead to severe health conditions. HCHO is a short-lived product derived by the oxida-
tion of volatile organic compounds (VOCs). Its sources are both natural (i.e., oxidation of VOCs emitted from plants), as well as anthropogenic (i.e., biomass burning, industrial-related emissions and road transport) (De Smedt et al., 2008; Chan et al., 2020). $NO_2$ is mainly produced by the oxidation of nitrogen monoxide (NO) and in most urban areas its sources include fossil fuel combustion, biomass burning, soil emissions and lightning (Lee et al., 1997; Zhang et al., 2003). Moreover, under cer-tain meteorological conditions, $NO_2$ may participate in the formation of secondary aerosols (Jang and Kamens, 2001). Given

the influence of $NO_2$, HCHO and aerosols on air quality and climate, it is of high environmental and research importance to monitor accurately and continuously their spatio-temporal distribution in the troposphere.

Multi-Axis Differential Optical Absorption Spectroscopy (MAX-DOAS) is a well-established ground-based passive remote sensing technique that received considerable attention during the past decades (Hönninger and Platt, 2002; Hönninger et al., 2004; Wagner et al., 2004; Wittrock et al., 2004; Frieß et al., 2006; Irie et al., 2008) and is nowadays widely used in many studies in order to simultaneously detect trace gases and aerosols mainly in the PBL and in the lowermost free troposphere (e.g., Clémer et al. (2010); Irie et al. (2011); Ma et al. (2013); Pinardi et al. (2013); Vlemmix et al. (2015a, b); Wang et al. (2017b); Chan et al. (2019) and references therein). Such trace gases include $NO_2$, HCHO, sulfur dioxide ($SO_2$), water vapour ($H_2O$), ozone ($O_3$), nitrous acid (HONO), iodine oxide (IO), glyoxal (CHOCHO) and bromine oxide (BrO). The MAX-DOAS measurement technique utilizes scattered sunlight in the ultraviolet (UV) and visible (VIS) part of the electromagnetic spectrum received from different elevation angles and the measured spectra are analyzed by Differential Optical Absorption Spectroscopy (DOAS) (Platt and Stutz, 2008) for the determination of the differential Slant Column Densities (dSCDs). Information about the vertical distribution of aerosols and trace gases can be retrieved from a single elevation sequence (i.e., spectra recorded at different elevation angles that belong to the same azimuthal direction) using suitable inversion algorithms. The products retrieved by the inversion algorithms include, among others, estimates of the profile shape, tropospheric Vertical Column Densities (VCDs) and near-surface concentrations.

Nowadays, there is a variety of such inversion algorithms for the retrieval of vertical profiles from MAX-DOAS measurements using different techniques. These algorithms are mainly separated into those that retrieve the profiles based on the optimal estimation method (OEM) (Rodgers, 2000) and into those that rely on a few parameters to characterize the atmospheric profile (parameterization approach). Both OEM-based and parameterized inversion algorithms have been tested and intercompared so far in many studies using either synthetic data (e.g., Frieß et al., 2019) or actual MAX-DOAS measurements, as for example, during the Cabauw Intercomparison of Nitrogen Dioxide Measuring Instruments 2 (CINDI-2) campaign (Wang et al., 2020; Tirpitz et al., 2021). Here, we use two of the already tested inversion algorithms to analyze MAX-DOAS measurements conducted at Thessaloniki, Greece, for the retrieval of aerosol, $NO_2$ and HCHO vertical profiles and column densities. These algorithms are the Mexican MAX-DOAS Fit (MMF) v2020_04 (Friedrich et al., 2019) and the Mainz Profile Algorithm (MAPA) v0.98 (Beirle et al., 2019). The former is based on the OEM, while the latter follows the parameterization approach and are both adopted by the Fiducial Reference Measurements for Ground-Based DOAS Air-Quality Observations (FRM$_4$DOAS) project (https://frm4doas.aeronomie.be/, last access: 05 March 2021). In this work we evaluate the performance of the two algorithms, we validate their results with reference datasets and we investigate the effect of applying different flagging schemes to the retrieved products. Additionally, by using two independent inversion algorithms, we aim at producing a reference MAX-DOAS dataset of higher quality for further research activities in Thessaloniki (e.g., validation of satellite-retrieved tropospheric products). Thessaloniki is also part of the FRM$_4$DOAS project, which aims at the development of the first central processing system for MAX-DOAS observations. Even though the measured spectra are regularly submitted and analyzed on a near-real-time basis, in this work both MMF and MAPA runs are performed offline in order to obtain more flexibility in the analysis and also to investigate and optimize the retrieval settings particularly for Thessaloniki.

The article is structured as follows. In Sect. 2 the instrumentation, the MAX-DOAS retrieval settings and a brief description of the profiling algorithms are reported, along with the methodology used in this analysis. In Sect. 3 we present the results of the comparison between different products retrieved by MMF and MAPA. In Sect. 4 the validation results of the retrieved products with ancillary data are presented and in Sect. 5 the main conclusions of this article are summarized.

## 2    Data and Methodology

### 2.1    Instrumentation

A 2D MAX-DOAS system (Phaethon) operates regularly on the rooftop (20 m above ground) of the Physics Department building of the Aristotle University of Thessaloniki (40.634$^o$ N, 22.956$^o$ E), about 60 m above sea level. The measurement site is located near the city center of Thessaloniki (Figure 1). The prototype system was developed in 2006 at the Laboratory of Atmospheric Physics (LAP) (Kouremeti et al., 2008, 2013) and has been upgraded ever since for the retrieval of tropo-
spheric $NO_2$ VCDs (Drosoglou et al., 2017, 2018) and total ozone columns (Gkertsi et al., 2018). The current version of the system comprises a single channel ultra-low stray-light AvaSpec-ULS2048x64-EVO ($f = 75$ mm) spectrometer by Avantes, the entrance optics and a two-axes tracker. The spectrometer's detector is a back-thinned Hamamatsu charge-coupled device (CCD) array of 2048 pixels with Signal to Noise Ratio (SNR) 450:1 for a single measurement at full signal. The spectrometer covers the spectral range 280 – 539 nm and uses a 50 $\mu$m wide entrance slit. Mercury discharge lamp spectra were recorded to
determine the instrument's slit function and the spectral resolution was found $\sim$ 0.55 nm full width at half maximum (FWHM) at 436 nm. The spectrometer is positioned inside a thermally isolated box, where the temperature is maintained at +10 $^o$C using a thermoelectric Peltier system. The entrance optics are mounted on a two-axes tracker with two stepper motors controlling the azimuth viewing angle ($0^o \leq \phi \leq 360^o$) and the elevation viewing angle ($0^o \leq \alpha \leq 90^o$) with pointing resolution of 0.125$^o$, allowing both direct-sun and off-axis observations. A third motor rotates a filter-wheel of 8 positions with different optical
components (diffuser, attenuation and band-pass filters), used for the measurement of direct-sun and scattered radiation spectra and an opaque position for the measurement of the dark signal. The instrument operates automatically and is controlled by a custom-made software, developed at LAP. The entrance comprises also a telescope with a plano-convex lens that focuses the collected solar radiation onto one end of an optical fiber. The system's field of view (FOV) was characterized using a distant light source and was found $\sim$ 1$^o$. Simultaneous azimuth and elevation angle calibration is regularly performed by sighting the
sun, so no horizon scans are necessary for the elevation angle calibration.

A routine MAX-DOAS measurement cycle starts by orienting the optics at a certain azimuth viewing direction followed by the measurement of scattered radiation spectra at the elevation angles: 90 (zenith), 30, 15, 12, 10, 8, 6, 5, 4, 3, 2 and 1$^o$ in this order. For this study, the system was configured to measure at four consecutive azimuth angles of 142, 185, 220 and 255$^o$, illustrated in Figure 2 with arrows of different colors. Based on the intensity of the measured spectra during an elevation
scan at 142$^o$ azimuth, the viewing direction of 1$^o$ elevation angle was found to be partly blocked by obstacles, such as trees and buildings in the campus. Thus, $\alpha = 1^o$ in this particular direction was excluded from the profiling analysis. In order to achieve high SNR values and to avoid saturated spectra, the number of scans of each individual measurement and the exposure

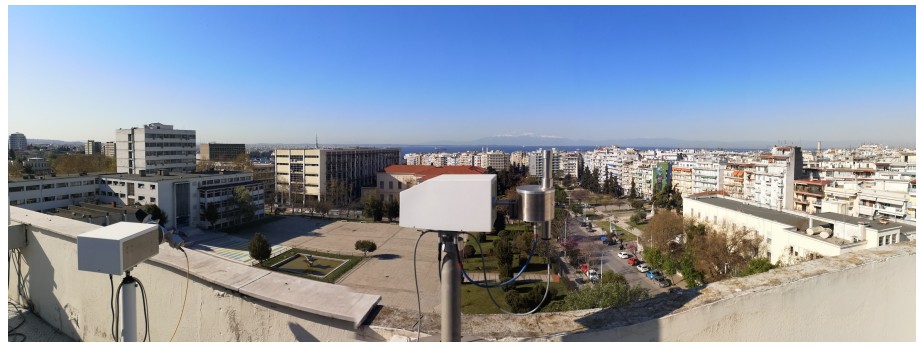

**Figure 1.** The Phaethon MAX-DOAS system in the middle and a panoramic view (East - South - West) of the measurement site.

time of the CCD are automatically adjusted by the operating software according to the received intensity by the detector. The integration time at each elevation angle is $\sim 60$ sec and a full measurement sequence for all azimuth directions lasts about one
120 hour.

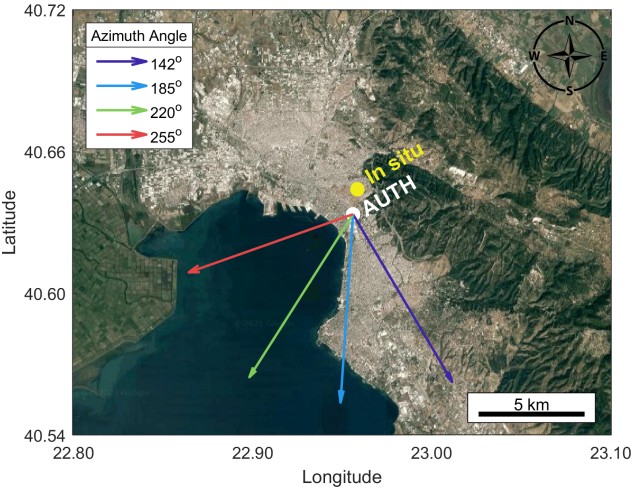

**Figure 2.** Location of the MAX-DOAS system (white dot) and the in situ $NO_2$ measurement site (yellow dot). The arrows in different colors represent the azimuth viewing directions, $\phi$, of the MAX-DOAS observations (i.e., purple: 142°, blue: 185°, green: 220° and red: 255°). The base map is taken from © Google Maps, https://www.google.com/maps/ (last access: 05 March 2021).

## 2.2 MAX-DOAS measurements and slant column retrieval settings

The primary retrieved product from the analysis of the measured MAX-DOAS spectra is the dSCD of several trace gases at different elevation angles. The dSCD of a trace gas at an elevation angle $\alpha$ ($dSCD_\alpha$) can be calculated as the difference between

the Slant Column Density, i.e., its concentration integrated along the light path ($SCD_\alpha$) and the SCD of a Fraunhofer reference spectrum (FRS), usually measured at the zenith ($SCD_{ref}$):

$$dSCD_\alpha = SCD_\alpha - SCD_{ref} \tag{1}$$

The MAX-DOAS spectra that are used in this study have been recorded for 1 year (from May 2020 through May 2021) and a zenith spectrum is selected as the FRS in order to account for the Fraunhofer lines and the stratospheric contribution of the absorbers (Hönninger et al., 2004). Since the system is scheduled to perform both direct-sun and MAX-DOAS observations during the day, the zenith spectra of two consecutive elevation sequences may have a large time difference (duration of the first sequence plus the duration of two direct-sun measurements). So, in this study, the zenith spectrum of each sequence was selected as the FRS for the DOAS-based retrieval of the collision-induced oxygen complex ($O_2$–$O_2$ or $O_4$) and the trace gas dSCDs and not the average or the time interpolated spectrum between the zenith spectra of the two consecutive sequences. The dSCDs of $O_4$ and trace gases are derived from the recorded spectra by applying the DOAS technique (Platt and Stutz, 2008), while the measured spectra are analyzed using the QDOAS (version 3.2, September 2017) spectral fitting software suite developed by BIRA-IASB (http://uv-vis.aeronomie.be/software/QDOAS/) (Danckaert et al., 2013). The retrieval settings are based on results from the CINDI-2 campaign (http://www.tropomi.eu/data-products/cindi-2/, last access: 05 March 2021) (e.g., Kreher et al., 2020), the Quality Assurance for Essential Climate Variables (QA4ECV) project (http://www.qa4ecv.eu/, last access: 05 March 2021) and the Network for Detection of Atmospheric Composition Change (NDACC) protocol for UV – VIS measurements (http://www.ndaccdemo.org/data/protocols/, last access: 05 March 2021). The spectral retrieval settings and the trace gas absorption cross sections that are included in the DOAS fit are listed in Table 1. The wavelength calibration of the measured spectra is achieved by shifting and stretching them against a highly resolved solar reference spectrum (Chance and Kurucz, 2010). Even though the spectrometer is operating in a temperature controlled environment, small diurnal temperature variations may occur. Thus, dark spectra are measured after each elevation sequence for all of the exposure times that were used during the sequence. This procedure might be time-consuming, but assures that the solar and dark spectra are measured at the same temperature. The dark spectra are then subtracted from the scattered radiation spectra prior to the DOAS analysis. Figure 3 shows a typical example of the DOAS analysis of a spectrum recorded on 9 July 2020 at 07:50 UTC at 3° elevation angle. During the whole period of study no apparent system-related issue or instrument degradation is observed.

## 2.3 Retrieval of the vertical profile

The retrieval of vertical profiles (extinction and concentration profiles for aerosols and trace gases, respectively) from MAX-DOAS measurements typically involves three major steps (Irie et al., 2011; Hendrick et al., 2014; Vlemmix et al., 2015b), independent of the retrieval approach. In the first step, the $O_4$ dSCDs and the trace gas dSCDs (in this case $NO_2$ and HCHO) are derived by applying the DOAS fitting technique to the measured spectra, as described in Sect. 2.2. Next, the $O_4$ dSCDs retrieved for each elevation angle of the same sequence are used as input to the algorithm for the retrieval of the aerosol

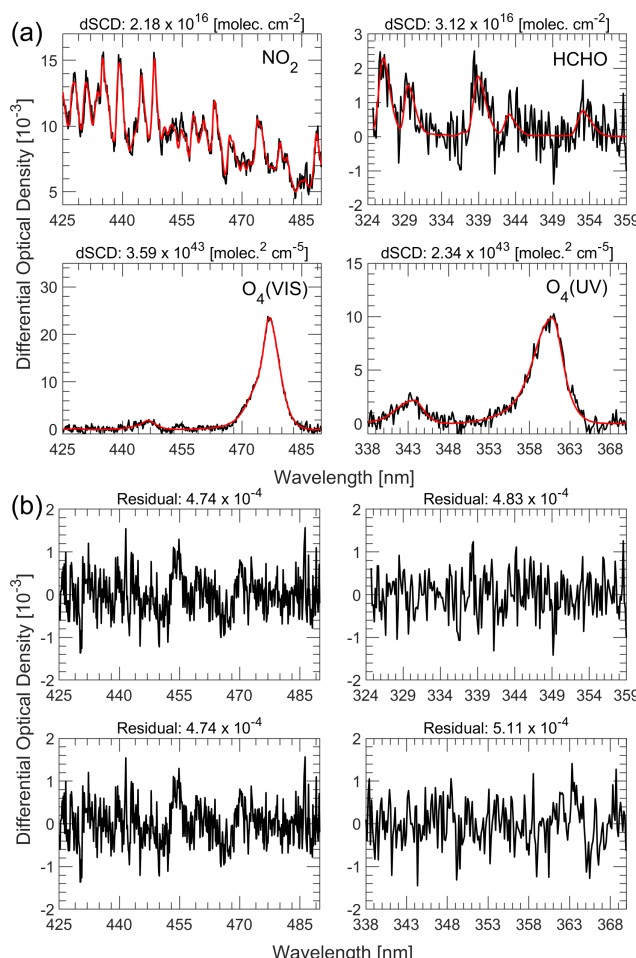

**Figure 3.** A typical example of the DOAS retrieval of $NO_2$, HCHO, $O_4$ (VIS) and $O_4$ (UV) dSCDs derived from a MAX-DOAS measurement on 9 July 2020 at 07:50 UTC (SZA $= 38.85^o$) at $3^o$ viewing elevation angle. The DOAS fits are presented in the figures of panel **a**. The black lines represent the measured spectra and red lines are the fitted $O_4$ and trace gas cross sections. The figures of panel **b** show the residual of the DOAS fits.

extinction vertical profile. In the end, the trace gas dSCDs are used as input to the algorithm for the retrieval of the trace gas vertical profile, along with the aerosol extinction profile, calculated in the previous step.

    As mentioned already, the profiling algorithms that have been developed so far and are commonly used within the MAX-DOAS community are either based on the OEM or follow the parameterization approach. How the $O_4$ and trace gas dSCDs are handled for the retrieval of the vertical profiles depends on each algorithm's approach. However, the principal idea of both

OEM and parameterized inversion algorithms is the same: A layered model atmosphere with defined parameters is assumed in a forward Radiative Transfer Model (RTM) and it is used in order to simulate the $O_4$ and trace gas dSCDs, taking into account the viewing geometry, i.e., the solar zenith angle (SZA), the elevation angle and the relative azimuth angle. The forward models

and how the dSCDs are simulated are described in Beirle et al. (2019) for MAPA and in Friedrich et al. (2019) for MMF. The extinction and concentration vertical profiles are derived by inverting the forward model, i.e., by finding the model parameters, for which the difference between the simulated and the measured dSCDs is minimized, based on a cost function.

**Table 1.** DOAS fit settings for $NO_2$, HCHO, $O_4$ (VIS) and $O_4$ (UV).

| Parameter | Data Source | trace gas | | |
|---|---|---|---|---|
| | | $NO_2$ and $O_4$ (VIS) | HCHO | $O_4$ (UV) |
| Spectral range | | 425 – 490 nm | 324.5 – 359 nm | 338 – 370 nm |
| $NO_2$ (298 K) | Vandaele et al. (1998), $I_0$-corrected (SCD in $10^{17}$ molecules cm$^{-2}$) | ✓ | ✓ | ✓ |
| $NO_2$ (220 K) | Vandaele et al. (1998), $I_0$-corrected (SCD in $10^{17}$ molecules cm$^{-2}$) | ✓ | | ✓ |
| $O_3$ (223 K) | Serdyuchenko et al. (2014), $I_0$-corrected (SCD in $10^{20}$ molecules cm$^{-2}$) | ✓ | ✓ | ✓ |
| $O_3$ (243 K) | Serdyuchenko et al. (2014), $I_0$-corrected (SCD in $10^{20}$ molecules cm$^{-2}$) | | ✓ | ✓ |
| $O_4$ (293 K) | Thalman and Volkamer (2013) | ✓ | ✓ | ✓ |
| BrO (223 K) | Fleischmann et al. (2004) | | ✓ | ✓ |
| HCHO (297 K) | Meller and Moortgat (2000) | | ✓ | ✓ |
| $H_2O$ (296 K) | HITEMP, Rothman et al. (2010) | ✓ | | |
| Ring | Ring spectra calculated by QDOAS according to Chance and Spurr (1997) | ✓ | ✓ | ✓ |
| Polynomial degree | | 5 | 5 | 5 |
| Intensity offset | | Constant | Order 1 | Constant |
| Wavelength Calibration | Based on a high-resolution solar reference spectrum (Chance and Kurucz, 2010) | | | |

## 2.4 MMF

The Mexican MAX-DOAS Fit (MMF) v2020_04 (Friedrich et al., 2019) is an OEM-based profiling algorithm that relies on online RTM simulations using VLIDORT version 2.7 (Spurr, 2006) as forward model. The input parameters for each atmospheric layer are calculated from temperature and pressure profiles, the trace gas concentration in each layer and the aerosol properties. The aerosol properties, which are the same for all layers, are the single scattering albedo (SSA) and the asymmetry parameter (using the Henyey–Greenstein phase function, Henyey and Greenstein (1941), to calculate the phase function moments). Furthermore, the wavelength of the retrieval and the surface albedo need to be specified as additional input parameters. The retrieval algorithm comprises an aerosol extinction profile retrieval and a trace gas profile retrieval. The former constrains the aerosol extinction profile in the forward model of the trace gas retrieval. The inversion uses constrained damped

least-square fitting with an optimal estimation regularization. In the used version, both the a priori and the covariance matrix are constructed. More details about the a priori settings and the input parameters can be found in Sect. 2.6. The retrieval algorithm provides the aerosol extinction profiles, trace gas partial column profiles, their integrated quantities, the corresponding noise and smoothing errors, as well as the averaging kernel, the degrees of freedom and a quality flag of the retrieval. The quality flagging system of MMF is based on the convergence of the algorithm, the root mean square of the difference between measured and simulated dSCDs, the reported degrees of freedom and the stability of the retrieval.

## 2.5 MAPA

The Mainz Profile Algorithm (MAPA) v0.98 (Beirle et al., 2019) is a profiling algorithm developed by the Max Planck Institute for Chemistry (MPIC) that is based on a parameterization approach. MAPA does not rely on online RTM simulations, but its forward model is provided as pre-calculated differential Air Mass Factor (dAMF) look-up tables (LUTs) at multiple wavelengths. These LUTs have been calculated offline by a full spherical RTM, McArtim (Deutschmann et al., 2011), following a backward Monte Carlo approach. Just like MMF, MAPA is based on a two-step process in order to retrieve the aerosol and trace gas vertical profiles. It uses three main parameters to characterize the atmospheric profile: The column parameter, $c$ (i.e., AOD for aerosols and VCD for trace gases), the layer height, $h$ and the shape parameter, $s$. Additionally, a fourth optional parameter can be included, the $O_4$ scaling factor, which was initially introduced by Wagner et al. (2009) in order to achieve agreement between the measured dSCDs and the forward model simulations. Unlike MMF, MAPA is not based on the OEM, so no a priori assumption of the vertical profile is required. In some cases this can be an advantage since a priori information and constraints are usually difficult to estimate. MAPA also provides a detailed flagging algorithm, that is based on thresholding techniques applied to different parameters, in order to evaluate whether the retrieved profile can be trusted. By default (and within this study), the flags are identical for the species retrieved in the UV and VIS spectral range. The flags that are defined in MAPA v0.98 are mainly based on the agreement between the measured and modeled dSCDs, the consistency of the derived Monte Carlo parameters and the shape of the profile. More details about MAPA and its flagging algorithm can be found in Beirle et al. (2019).

## 2.6 Input parameters and settings

During MAPA calculations, depending on the aerosol or trace gas retrieval, a LUT corresponding to the central wavelength of the $O_4$ or trace gas fitting window is selected (i.e., 360 nm for $O_4$ in the UV, 343 nm for HCHO, 460 nm for $NO_2$ and 477 nm for $O_4$ in the VIS). These wavelengths are also used in the RTM simulations of MMF. For the calculation of the dAMF LUTs, MAPA's radiative transfer simulations were performed with a typical fixed set of parameters for all wavelengths (Beirle et al., 2019), which can describe the majority of all potential measurement sites. MMF, on the other hand, relies on online RTM simulations and so the aerosol and surface parameters can be adjusted to the most suitable values. In this study, the aerosol optical properties that are used as input for the simulations of MMF are based on 15 years climatological data measured by a co-located CIMEL sun-photometer. Figure 4 shows the frequency distribution of the Ångström exponent, AOD, asymmetry

factor and SSA in Thessaloniki, while their values that are used as input to each inversion algorithm are listed in Table 2. Discrepancies between MMF and MAPA due to small differences in these selected parameters are expected to be minor.

**Table 2.** The RTM settings that were used in MMF and MAPA for Thessaloniki.

| Parameter | Inversion algorithm | |
|---|---|---|
| | MAPA | MMF |
| Aerosol single-scattering albedo | 0.95 | 0.95 |
| Aerosol asymmetry parameter | 0.68 | 0.69 |
| Surface albedo | 0.05 | 0.06 |
| Ångström exponent | 1.4 | 1.4 |

MMF requires a priori profile and covariance matrix information for the profile retrievals. The "a priori" term represents
knowledge of the true state before the measurement is performed. However, the true shape of the trace gas vertical profiles at Thessaloniki is generally not known, while the true state of the aerosol profiles is known only in certain cases during the period of study. Thus, the retrieval is based on constructed exponentially decreasing a priori profiles with scale height of 1 km, which are considered a reasonable estimate of the true profiles. Since no covariance matrix information is available, the covariance matrix is also constructed from the a priori profile. The AOD as well as the trace gas VCDs in Thessaloniki vary
substantially throughout the year. In order to take into account the annual variability, we use the square of 50% of the a priori on the diagonal elements of the covariance matrix for aerosols and 100% for $NO_2$ and HCHO. The loose constraint of the latter is due to the higher variability of the trace gas vertical columns over the course of the year. Both for aerosols and trace gases, the off-axis elements of the covariance matrix were constructed by assuming a Gaussian function with correlation length of 200 m, as described in Clémer et al. (2010). Additionally, based on empirical tests, the progress of the convergence is faster
when using an a priori VCD or AOD below the true value for reasons that are yet not identified. Thus, the a priori AODs were set to 0.25 and 0.15 for the aerosol retrievals at 360 and 477 nm, respectively. For the trace gas retrievals we have used a priori VCDs of $4 \times 10^{15}$ and $6 \times 10^{15}$ molecules cm$^{-2}$ for $NO_2$ and HCHO, respectively, based on data derived from the MAX-DOAS by applying the geometrical approximation (Hönninger et al., 2004) to the dSCDs measured at 30º and 15º elevation angles. The LUTs used in MAPA cover the following ranges: 0 – 5 for the AOD, 0.02 – 5 km for the layer height and 0.2 – 1.8 for the
profile shape parameter (Beirle et al., 2019).

The temperature and pressure vertical profiles that are used as input in this study are identical for both MMF and MAPA. We have used climatological profiles for Thessaloniki produced by MPIC that are based on ∼ 16 years re-analysis data from the European Centre for Medium-Range Weather Forecasts (ECMWF). The temperature and pressure profiles are interpolated to the day and time of each elevation sequence. We have also tried to use temperature and pressure profiles measured by
radiosondes, launched on a daily basis at Thessaloniki Airport (∼ 13 km away from the measurement site), as input, but since

no major effect is observed on the retrieved products, these results are not presented. Both algorithms are configured to export the retrieved vertical profiles to the same output grid ranging from the ground up to 4 km with 200 m vertical resolution.

As already mentioned, the recorded spectra are also analyzed by a central processing system in the frame of the FRM$_4$DOAS project. The analysis is carried out using default values of several parameters, which are reasonable for all potential measurement sites, while in this study we try to optimize the performance of MMF and MAPA particularly for Thessaloniki (see discussion above). In the FRM$_4$DOAS analysis a time interpolated spectrum between the zenith spectra of two consecutive elevation sequences is used as the FRS for the dSCDs retrievals. Thus, the dSCDs that are used for the retrieval of the vertical profiles are slightly different than those used in the current study. The default FRM$_4$DOAS settings include: SSA of 0.92 and asymmetry factor of 0.68. Yet, such small differences should have a negligible effect on the retrieved vertical profiles. The Ångström exponent is set to 1 and the same a priori aerosol extinction vertical profile (AOD of 0.18) is used for the retrievals both in the UV and VIS spectral range. The covariance matrices are constructed from the a priori profiles, but the square of 50% of the a priori is used on the diagonal elements of the covariance matrices for all species. Currently, the partly blocked elevation angle of 1° at 142° (Sect. 2.1) is not excluded from the analysis. MAPA retrievals are performed using three different O$_4$ scaling factors (i.e., 0.8, 1.0 and a variable scaling factor). In order to investigate further the effect of the O$_4$ scaling factor (see Appendix A) in this study we include an extra value of 0.9.

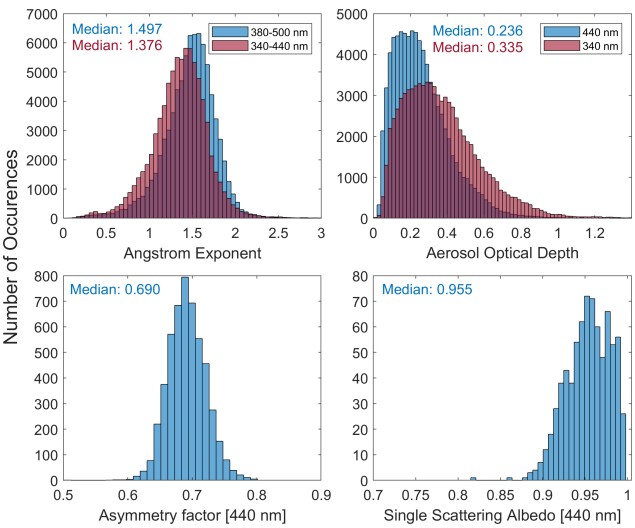

**Figure 4.** Frequency distribution of the Ångström exponent, AOD, asymmetry factor and SSA measured by a CIMEL sun-photometer in Thessaloniki for the period 2005 – 2021.

## 2.7 Ancillary data

This section describes briefly the supporting instruments that are used in this study for comparison and validation of MAX-DOAS derived products. The ancillary data include measurements of a CIMEL sun-photometer, a Brewer spectrophotometer,

an aerosol lidar system and an in situ $NO_2$ monitoring station. Except for the in situ $NO_2$, all remote sensing instruments that
are used in this study (i.e., the MAX-DOAS, CIMEL, Brewer and lidar) are located at the same measurement site, about 60 m above sea level. The effect of the different viewing geometries and the retrieval techniques that each system utilizes are discussed at the corresponding following sections.

### 2.7.1 CIMEL sun-photometer

Since 2003 a Sun Sky photometer (CIMEL) provides spectral measurements of the AOD at Thessaloniki as part of the NASA's
Aerosol Robotic Network (AERONET) (https://aeronet.gsfc.nasa.gov/). CIMEL is an automated, well-calibrated scanning filter radiometer specifically developed for the retrieval of the AOD at 7 wavelengths (i.e., 340, 380, 440, 500, 670, 870 and 1020 nm) by using direct-sun observations. The technical specifications of the instrument are given in Holben et al. (1998). The instrument is calibrated regularly following the procedures and the guidelines of AERONET. The AERONET database provides three distinct levels for data quality. Level 1.0 is defined as pre-screened data (i.e., no quality assurance criteria are
applied). The Version 3 (Sinyuk et al., 2020) of Level 1.5 represents near-real-time automatic cloud screened data, while Level 2.0 applies additional pre- and post-field calibrations. In this paper, we use the AERONET Level 1.5 data, since the Level 2.0 data for the period of study is not yet published. In order to compare with the AOD retrieved by the MAX-DOAS, the AODs at 360 and 477 nm have been calculated using the Ångström exponent between the standard spectral bands of the instrument.

### 2.7.2 Brewer spectrophotometer

The Brewer spectrophotometer with serial number 086 (B086) is a double monochromator that performs spectrally resolved measurements of the direct and global solar irradiance at Thessaloniki since 1993 (Bais et al., 1996; Fountoulakis et al., 2016). The wavelength range of B086 is 290 – 365 nm and its spectral resolution is 0.55 nm at full width at half maximum (FWHM). The wavelength calibration is performed by scanning the emission lines of spectral discharge lamps, while maintenance of the absolute calibration is achieved by regularly scanning the spectral irradiance of a calibrated 1000-W quartz–halogen tungsten
lamp (Garane et al., 2006).

Although the Brewer's initial purpose was the retrieval of total ozone columns, research activities have shown that the spectral AOD can be calculated from direct irradiance measurements by following two main approaches: The first is based on the absolute calibration of the direct-sun spectra measured by the Brewer (Kazadzis et al., 2005), while the second uses the Langley extrapolation method (relative calibration) (Gröbner and Meleti, 2004). In both cases the spectral AOD is calculated
as the residual optical depth after subtracting from the total atmospheric optical depth the optical depths due to molecular scattering and the $O_3$ and $SO_2$ absorption (Kazadzis et al., 2007). Since 1997, the direct solar irradiance spectra measured by the Brewer are calibrated (Bais, 1997), so in this study we use the former approach (i.e., absolute calibration) for the retrieval of the spectral AOD. In order to compare with the AOD retrieved by the MAX-DOAS, the AOD at 477 nm is calculated using climatological monthly mean values of the extinction Ångström exponent derived from measurements of the CIMEL
sun-photometer in Thessaloniki. Details on the procedure of Brewer's direct solar irradiance spectra absolute calibration, as

well as the spectral AOD retrieval methodology can be found in Bais (1997); Kazadzis et al. (2005, 2007); Fountoulakis et al. (2019).

### 2.7.3 Lidar

Thessaloniki is a member station of the European Lidar Aerosol Network (EARLINET, https://www.earlinet.org) since 2000, providing regular aerosol profile measurements, following the EARLINET's schedule (Monday morning, Monday and Thursday evening), during extreme events and at satellite overpasses (e.g., AEOLUS, OMI).

THEssaloniki LIdar SYStem (THELISYS) is a multi-wavelength Raman/depolarization lidar system, which has been gradually upgraded regarding its operational wavelengths and the detection configuration. All the quality standards, established within EARLINET, are followed in order to assure the high quality of the THELISYS products, which are publicly available in the EARLINET database (https://www.earlinet.org/index.php?id=125). A detailed description of THELISYS technical specifications and algorithm can be found in Voudouri et al. (2020).

The final products derived from the raw lidar data processing are: the aerosol backscatter coefficient at 355, 532 and 1064 nm, the aerosol extinction coefficient at 355 and 532 nm and the linear particle/volume depolarization ratio at 532 nm. During the day, the data acquisition is limited to the signals that arise from the elastic scattering of the laser beam by the air molecules and the atmospheric aerosol. The Klett-Fernald algorithm in backward integration mode is applied (Klett, 1981) and the backscatter coefficient profiles are produced. Constant a priori climatological values of the ratio between the extinction and the backscatter coefficient (Lidar Ratio) were assumed in this daytime method. Values of 60, 50 and 40 were used for 355, 532 and 1064 nm, respectively, given the atmospheric situations that occur over Thessaloniki (Voudouri et al., 2020). The resulting uncertainties are discussed in depth by Böckmann et al. (2004) and can be as high as 50% if there is no information about the actual Lidar Ratio, during extreme atmospheric conditions.

Another source of uncertainty during the lidar signals processing is the system's overlap function, which determines the altitude, above which a profile contains trustworthy values. In our analysis, the correction is not available for the daytime retrievals. Thus, an overlap function from the previous nighttime measurement or a mean overlap profile is applied. The starting height is set to the full overlap height (approximately 0.6 km), assuming height-independent backscatter below 0.6 km, equal to the backscatter measured at this height, to account for both the incomplete overlap within the lidar profile and atmospheric variability in the lowermost tropospheric part. This overlap effect generally introduces uncertainties in the calculation of the columnar products (e.g., AOD). However, long term comparisons (Siomos et al., 2018) have shown similar decreasing trends of the AOD at 355 nm between the EARLINET and the AERONET datasets (-23.2% and -22.3% per decade, respectively). The AODs at 355nm measured by the lidar have also been compared with the Brewer's retrievals, showing a generally good correlation of 0.7 (Voudouri et al., 2017).

### 2.7.4 In situ

Near-surface concentrations of different air pollutants, including $NO_2$, NO, $SO_2$, CO and $O_3$ are measured in Thessaloniki by in situ instruments as part of the Network for Air Quality Monitoring of the Municipality of Thessaloniki. $NO_2$ is being

monitored by chemiluminescence detectors that are mainly distributed around the city center. Most of the network stations are
315 installed very close to the ground (sampling inlet at $\sim 3$ m) and are strongly affected by local traffic emissions. In this study, we
use hourly mean (which is the highest available temporal resolution) in situ $NO_2$ concentrations measured at the "Eftapyrgion"
site (40.644º N, 22.957º E, 174 m a.s.l.), which is located in an urban background area at a distance of $\sim 1.2$ km from the
MAX-DOAS system to the North (Figure 2). The in situ measurements, spanning from May 2020 to March 2021, are used in
order to validate the MAX-DOAS-derived $NO_2$ near-surface concentrations. Even though this site is located opposite to the
320 MAX-DOAS system's azimuth viewing directions, it has been selected because the vertical and horizontal displacement of the
two instruments is small, but also because it is the only site of the network almost unaffected by local traffic emissions and
therefore can be considered more representative of the average $NO_2$ concentrations in the local boundary layer.

## 3   Results and discussion

In this section, we present results of the trace gas and aerosol quantities retrieved by the two inversion algorithms. We in-
325 tercompare the dSCDs simulated by the forward models, the integrated columns (i.e., VCDs and AODs for trace gases and
aerosols, respectively), the near-surface concentrations and the seasonal mean vertical profiles between MMF and MAPA.
Since MAPA is based on a parameterization approach, no information about averaging kernels is provided; hence, results on
averaging kernels are presented only for MMF.

The MAX-DOAS system operates at a site where the northern viewing directions are blocked by buildings of the campus
and the city, so the system is configured to perform sequences of elevation scans at azimuth directions in the southern sector,
as illustrated in Figure 2. As a result, scattered radiation spectra may be measured during the day at azimuths close to the
solar azimuth angle. In such cases, RTM simulations might face difficulties in calculating properly the dAMF due to increased
aerosol forward scattering, leading usually to underestimation of the true dAMF. For small scattering angles the uncertainties
caused by the incorrect description of the phase function can also become important and the results for such viewing geometries
should be treated with caution. Therefore, the elevation sequences measured at azimuth angles relative to the sun of less than
5º are excluded from the analysis. In addition, the elevation sequences, for which the retrieved AOD from the MAX-DOAS
inversion algorithms is greater than 1.5 are filtered-out, since such high aerosol loads are unrealistic for Thessaloniki (Figure
4). Negative columns can occur in the trace gas retrievals of MAPA within the Monte Carlo ensemble and they are by default
not removed, but this is not possible for MMF retrievals since, in its current version, MMF operates in logarithmic state vector
space. For $NO_2$, no valid negative columns are retrieved, but for HCHO, MAPA reports negative columns for $\sim 8.5\%$ of the
valid data. In order to compare meaningful results between the two algorithms, the negative columns are removed from the
initial dataset.

The individual flagging schemes of MMF and MAPA have been discussed elsewhere. Based on synthetic data, Frieß et al.
(2019) reported that the quality flagging criteria of MAPA might be too strict, since a large fraction of data was flagged as
invalid, even though the algorithm successfully removed almost all outliers. In our study, MAPA flags a larger fraction of data
as invalid, compared to MMF, for all the retrieved species. The percentage of the valid data flagged by MAPA and MMF

(individually and combined) is presented in Table 3. Since in MAPA retrievals no a priori constraints are used, more strict flagging needs to be applied for retrieved dSCDs that are characterized by large uncertainties (e.g., due to larger fit error or the effects of clouds). Especially for HCHO, the apparent worse performance of MAPA could be explained by the lower SNR in the UV, along with the higher HCHO profile height compared to $NO_2$ (see discussion in Sect. 3.5) and the decreasing sensitivity towards higher altitudes. The retrieval results are sensitive to the validity flagging approach, which is further investigated in the next section. No cloud filtering is applied to the data prior to the profiling analysis. Neither MMF nor MAPA include a direct cloud flagging system. However, some flags that are included in the flagging algorithms of MMF and MAPA are sensitive to clouds. Hence, in order to achieve retrievals of high quality and to ensure that the MAX-DOAS measurements performed under broken cloud conditions are filtered-out, an elevation sequence is considered valid as long as it is flagged as valid by both MMF and MAPA. This is the default flagging scheme for $NO_2$, HCHO and AOD at 477 nm and all the results shown in the next sections follow this flagging approach unless stated otherwise. For AOD at 360 nm the flags reported by MAPA are considered as default, since this approach performs better when comparing the MAX-DOAS results with other reference instruments (Sect. 4), although the reason for this behavior has not yet been identified. Also, since the issue for selecting the optimum $O_4$ scaling factor remains unresolved (Beirle et al., 2019; Wagner et al., 2019, 2021), we let MAPA determine an optimum $O_4$ scaling factor (variable) for each elevation sequence and this option is selected as the default for the retrievals.

It should be noted that in the following sections an Orthogonal Distance Regression (ODR or bivariate least-squares) has been used instead of an Ordinary Linear Regression (OLR or standard least-squares) for the comparison of the retrieved products derived by MMF and MAPA, in order to treat equally the two algorithms since none of them depends on the other. The discrepancies in the regression slopes and intercepts arising in the OLR when comparing independent variables, and the appropriateness of ODR, are discussed in Cantrell (2008). The ODR results are also sensitive to the assumed errors of the two variables. The uncertainty contained in the MAX-DOAS measurements may be difficult to assess, but, since both MMF and MAPA retrievals are based on the same input data, the associated errors are assumed the same and equal to the mean error provided by MMF and MAPA for each data point.

**Table 3.** The fraction of the data (%) that are flagged as valid by MMF and MAPA (individually and combined) for each species.

| Species | Inversion Algorithm | | Combined flagging |
|---|---|---|---|
| | MAPA | MMF | |
| $NO_2$ | 29.0 | 62.4 | 23.9 |
| HCHO | 18.0 | 82.6 | 16.8 |
| Aerosols (VIS) | 47.6 | 57.4 | 33.4 |
| Aerosols (UV) | 38.3 | 54.8 | 24.8 |

## 3.1 Simulated dSCDs

In this section we evaluate the performance of the forward models of MMF and MAPA by intercomparing the simulated trace gas dSCDs of the four species for the entire period. Also, we assess their ability to successfully simulate the slant column densities under different atmospheric (pollution and meteorological) conditions and viewing geometries by comparing the modeled with the measured dSCDs (Figure 5). Each row corresponds to a different trace gas, with the left column presenting the intercomparison results of the modeled dSCDs, while the middle and right columns show the comparison results between the measured dSCDs and the dSCDs simulated by MAPA and MMF, respectively. The data points are colored by the elevation angle and hotter colors represent dSCDs close to the horizon. A generally better performance of both algorithms is observed for the species retrieved in the VIS range compared to those retrieved in the UV. The modeled slant columns agree well, with Pearson's correlation coefficients and slopes close to unity ($R = 0.999$, Slope $= 1.008$ for $NO_2$ and $R = 0.998$, Slope $= 1.006$ for $O_4$ VIS). Additionally, the simulated dSCDs are in good agreement with the measured dSCDs, which is a good indicator for successful profile retrievals. In the case of $O_4$ (UV), even though the slope and correlation coefficient are similar to $O_4$ (VIS), a larger scatter is evident, while for HCHO larger deviations from unity in the slopes and correlation coefficients are observed, especially at higher elevation angles. This can probably be explained by the increased noise in the UV spectra compared to the VIS range and also due to the fact that at higher elevation angles the measured differential optical densities are very low, reaching the spectrometer's detection limit. For aerosols in both spectral ranges, discrepancies between the simulated dSCDs of MMF and MAPA may arise due to the variable $O_4$ scaling factor that is included in MAPA retrievals. This could also be the main driver of the positive bias for low elevation angles that is found in MMF's $O_4$ dSCDs (especially in the UV), while the results of MAPA are less affected.

## 3.2 Averaging Kernels

The averaging kernels (AVKs) of a profile retrieval describe the sensitivity of the retrieved state to the true atmospheric state for each altitude layer. The degrees of freedom (DoF) are mathematically derived as the trace (or sum of the diagonal elements) of the AVK matrix and quantify the number of independent pieces of information gained from the measurements compared to the a priori knowledge (Rodgers, 2000). Both the AVKs and the DoF can be used to characterize the quality of the retrieved profile. Since only OEM-based inversion algorithms are capable of providing AVKs, the results shown here are derived only by MMF. Figure 6 shows a typical example of the calculated AVKs for each of the retrieved species, including their corresponding DoF. The median DoF retrieved by MMF are: $3.13 \pm 0.32$ for $NO_2$, $2.22 \pm 0.34$ for HCHO, $2.73 \pm 0.28$ for aerosols in the VIS and $2.02 \pm 0.32$ for aerosols in the UV. The averaging kernels illustrate that MAX-DOAS measurements are typically less sensitive for altitudes greater than $\sim 2$ km, as a result of the viewing geometry, and thus, altitudes greater than 3 km are not presented here. That means that the MAX-DOAS measurements under these viewing geometries and with the a priori profiles and covariance matrices used in this study (Sect. 2.6) are adequate for retrieving the extinction and concentration profiles only up to the lowermost $\sim 1.5 - 2$ km of the atmosphere with highest sensitivity closer to the ground. Also, since the photon path

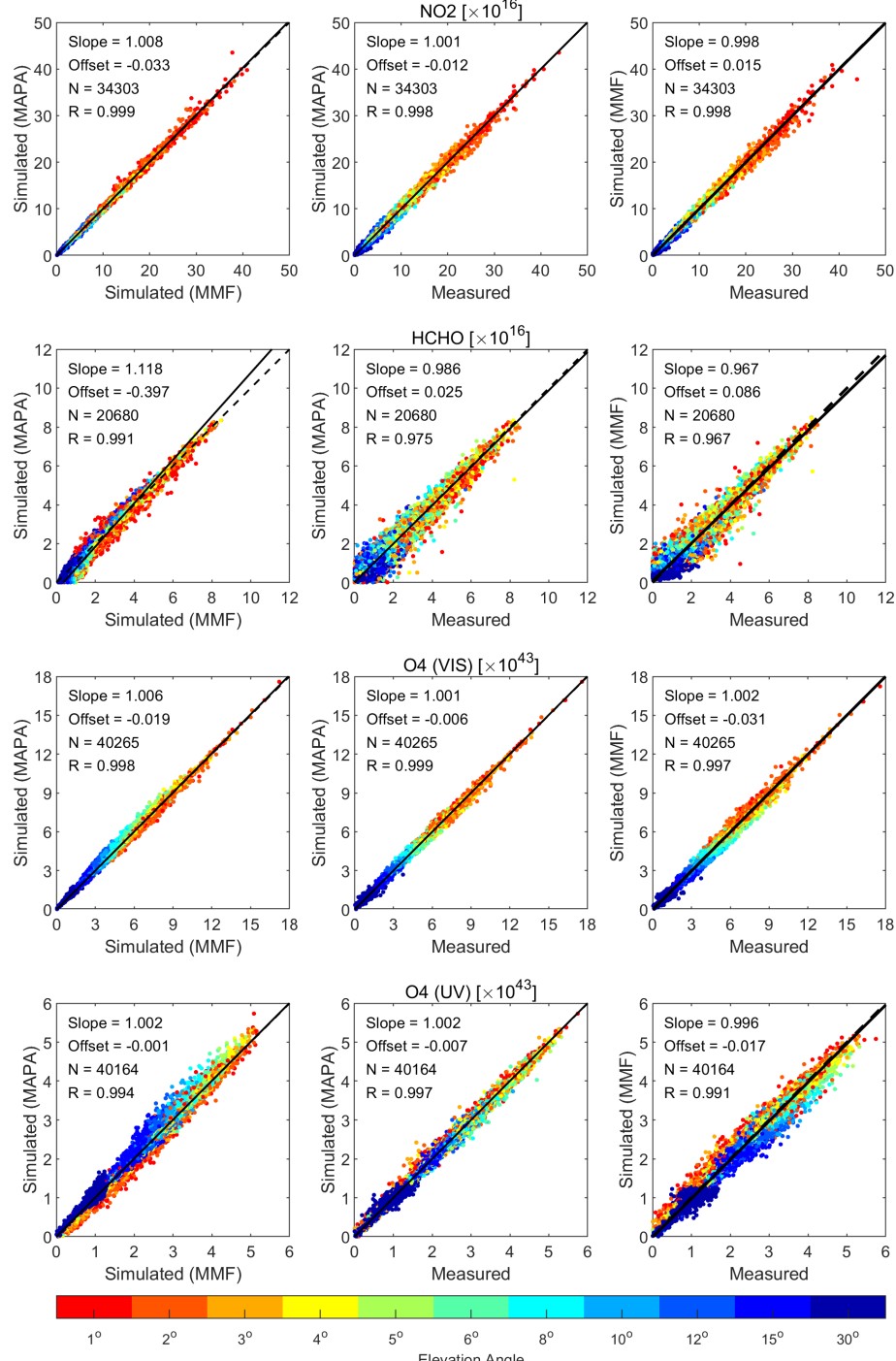

**Figure 5.** Intercomparison of the dSCDs simulated by MMF and MAPA (left column) and comparison of the dSCDs simulated by MAPA (center column) and MMF (right column) against the measured dSCDs. The elevation angles are denoted by different colors (see scale at the bottom).

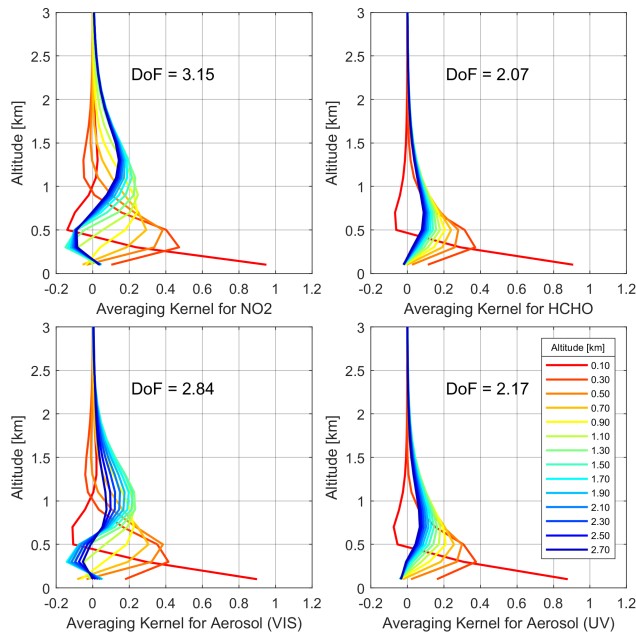

**Figure 6.** A typical example of the retrieved averaging kernels for different altitudes of each species. Hotter colors correspond to altitudes closer to the ground.

increases with wavelength, the MAX-DOAS technique shows higher sensitivity for the species retrieved in the VIS range than in the UV.

### 3.3 Integrated Columns

In the past, the trace gas VCDs measured by our MAX-DOAS systems have been derived by dividing the measured dSCDs, only at two elevation angles, $30^o$ and $15^o$, or the mean of the two, with appropriate dAMFs. The dAMFs have been calculated either following the geometrical approximation approach or by deploying RTM simulations taking into account the viewing geometry, the aerosol optical properties and the instrument's viewing direction relative to the sun (Drosoglou et al., 2017). However, in both cases, the actual trace gas profile has not been taken into consideration introducing, possibly, an additional

uncertainty to the measured VCD. This is the first time during the Phaethon's operation that the whole elevation sequence is used in order to retrieve the tropospheric VCDs more accurately. The comparison of the NO$_2$ and HCHO VCDs that are derived from the integration of the vertical profiles with the VCDs that are calculated using the geometrical approximation can be found in the Appendix B.

In Figure 7 the time series of the integrated columns of all retrieved species (i.e., AODs for aerosols and VCDs for trace

gases) are presented, as well as comparisons between MMF and MAPA. The statistics of the comparisons, i.e., slope ($S$), offset ($O$), number of points ($N$) and Pearson's correlation coefficient ($R$) are shown in different colors for each azimuth viewing direction. The text in black color represents the consolidated statistics for all azimuth directions. No clear azimuth

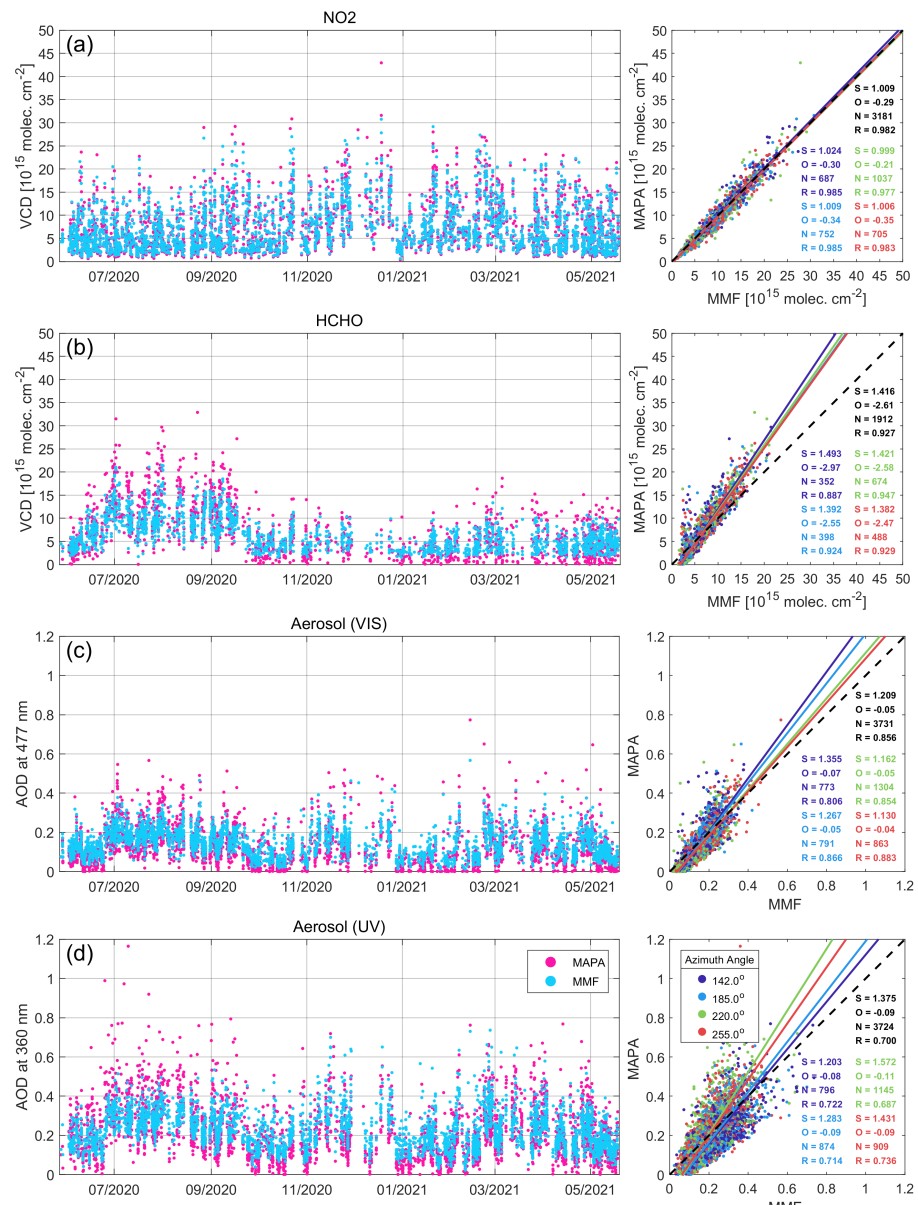

**Figure 7.** Time series and scatter plots of the integrated columns for all species retrieved by MMF and MAPA (panel **a** refers to NO₂, **b** to HCHO, **c** to aerosols in the visible range and **d** to aerosols in the UV). The parameters of the orthogonal distance regression, i.e., slope ($S$), offset ($O$), number of points ($N$) and Pearson's correlation coefficient ($R$) are shown in different colors for each azimuth viewing direction. The text in black color represents the consolidated statistics for all azimuth directions. The dashed black line represents the 1:1 line.

dependence of the retrieved columns is observed for the trace gases. However, for aerosols, especially in the UV, significant differences in the regression slopes appear for the different azimuths. It should be noted that in these comparisons a variable

$O_4$ scaling factor has been used for the MAPA retrievals, and since no scaling factor has been applied to the MMF retrievals, differences in the AODs between the two algorithms are expected. The number of elevation sequences for $220^o$ azimuth is always larger compared to the other azimuth directions, because the instrument was configured to record spectra only at this particular direction for approximately one month in the beginning of its operation. The comparison shows that the $NO_2$ VCDs derived by MMF and MAPA are in very good agreement, with slopes and correlation coefficients close to unity

(ranges: $0.999 \leq S \leq 1.024$ and $0.977 \leq R \leq 0.985$). Similar results were obtained for all azimuth directions with $S = 1.009$ and $R = 0.982$. In the case of HCHO, despite the good correlation ($R = 0.927$), notable deviations from unity in the slope are observed for all azimuth directions. MAPA systematically reports larger VCDs than MMF for higher HCHO concentrations, while the opposite behavior is observed for low HCHO loads, indicating that further investigation is required. This behavior could be explained by the increased spectral noise in the UV that leads to discrepancies between the HCHO dSCDs simulated

by the forward models of MMF and MAPA (see discussion in Sect. 3.1) and due to invariant a priori profile during the year. The sensitivity of the MAX-DOAS decreases with altitude and it is very limited at altitudes above $\sim 2.5$ km for the species measured in the VIS spectral range or even lower ($\sim 1.5$ km) for the species in the UV (Figure 6). For $NO_2$, this is generally not a problem since the total column is dominated by the concentration in the lower layers of the troposphere (see also discussion in Sect. 3.5). However, HCHO can be vertically extended at higher altitudes, where the sensitivity of the MAX-DOAS is low. In

the case of HCHO, MMF is more prone to result in the a priori profile, while MAPA retrievals become more unstable. Thus, the vertical profiles of MAPA are expected to have greater variability. Concerning aerosols, the comparison of the retrieved AODs reveals better agreement at 477 nm ($R = 0.856$) than at 360 nm ($R = 0.700$), with larger scatter and more outliers compared to the trace gas VCDs. As already mentioned, this is mainly attributed to the $O_4$ scaling factors that are used in MAPA retrievals. More details about the effect of the $O_4$ scaling factor on the retrieved AODs and the trace gas VCDs can be found in the

Appendix A.

### 3.4 Surface concentrations

The surface concentration is defined as the trace gas amount at ground level. However, the profile parameterization used within MAPA allows for the retrieval of lifted trace gas layers for a shape parameter greater than 1, which leads to a value of zero for the concentration at the surface. For these cases, the comparison with in situ measurements or surface concentrations retrieved by an OEM-based algorithm will be low-biased. Thus, in the following sections, the term "surface concentration" will refer to

445 the mean "near-surface concentration", i.e., the average concentration below 200 m for both MMF and MAPA, rather than the concentration directly at the ground. Figure 8 shows the time series of the near-surface $NO_2$ and HCHO concentrations derived by MMF and MAPA and the corresponding scatter plots. The comparisons of the surface values are similar to the comparisons of the tropospheric VCDs (shown in Figure 7) with Slope $= 1.118$, $R = 0.919$ for $NO_2$ and Slope $= 1.295$, $R = 0.855$ for

HCHO, but more outliers are present. In the case of HCHO, the surface concentrations derived for the $142^o$ azimuth show larger differences compared to the other directions, while this is not clear for $NO_2$. These discrepancies are possibly related to the fact that for this azimuth, the elevation angle of $1^o$ was not included in the analysis (see Sect. 2.1), which may have influenced the retrieved surface concentrations.

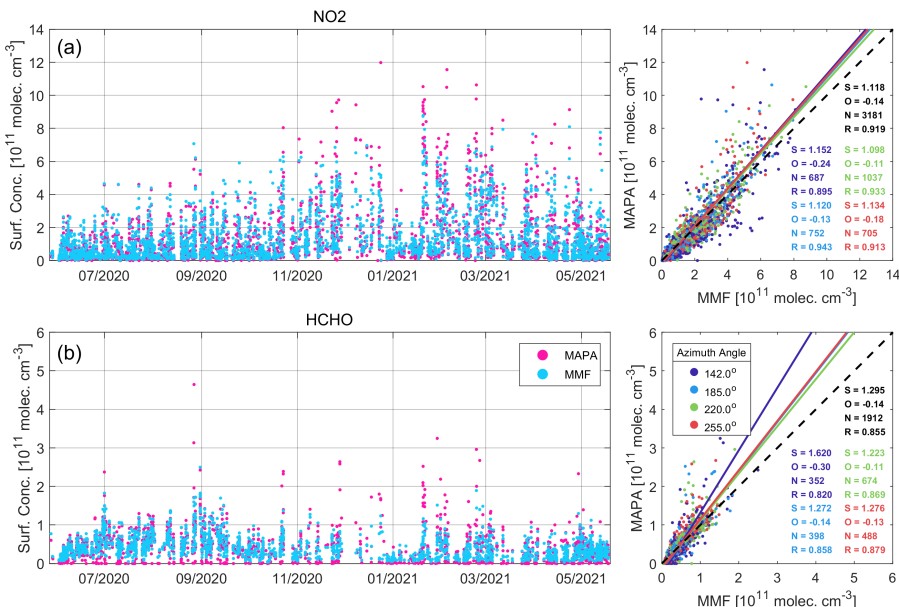

**Figure 8.** Time series and scatter plots of the near-surface concentrations of $NO_2$ (**a**) and HCHO (**b**) derived by MMF and MAPA. The parameters of the orthogonal distance regressions are presented as in Figure 7.

## 3.5 Seasonal mean vertical profiles

Figure 9 shows the seasonal mean $NO_2$, HCHO and aerosol extinction vertical profiles at Thessaloniki retrieved by MMF (cyan) and MAPA (magenta) during the 12 months considered in this study. Each row represents the vertical profiles of a specific species and each column corresponds to a different season. The shaded areas represent the standard deviation around the mean for each layer and they illustrate the seasonal variability of the vertical profiles. For $NO_2$ both algorithms report profiles that are decreasing with altitude for all seasons. Compared to MAPA, MMF reports slightly lower $NO_2$ concentrations below 1 km (yet within the range of variability) and slightly higher above 1 km. For all seasons, the variability of MMF's profiles between 1 and 2 km is larger, probably due to the increased contribution of the a priori profile under certain conditions (e.g., high aerosol load or fog close to the ground), where the sensitivity of the MAX-DOAS is lower. However, the seasonal mean profiles of both algorithms indicate that most of the $NO_2$ content lies within the first $\sim 500$ m. $NO_2$ originates mainly from direct, local emissions close to the ground (e.g., road transport emissions). Additionally, its lifetime in the PBL is short, typically a few hours depending on the season (e.g., Beirle et al., 2011; Liu et al., 2016) and as a result, higher $NO_2$ amounts are expected at lower altitudes in the troposphere. Both algorithms retrieve higher concentrations close to the surface during the cold period, which can be mainly attributed to enhanced $NO_2$ emissions near the ground (e.g., from domestic heating sources) and to reduced photolysis rates due to weaker solar radiation.

An opposite seasonal variation is observed for HCHO, with higher concentrations reported by both algorithms during summer (consistent with the VCDs, shown in Fig. 7). The profile shapes of MMF and MAPA agree reasonably well. In summer,

the larger retrieved concentrations are probably due to the increased emissions of VOCs, whose oxidation produces HCHO. According to Zyrichidou et al. (2019), biogenic emissions are expected to peak during summer, while the anthropogenic emissions do not show a clear seasonal variation in Thessaloniki. Therefore, the observed HCHO seasonality is mainly attributed to the enhanced biogenic emissions from vegetation in summer. VOCs are generally well mixed and have longer life times (Chan et al., 2019), hence, larger HCHO amounts are expected at higher altitudes during the warm season. MMF's profiles peak at a slightly higher altitude ($\sim 800$ m) than MAPA's ($\sim 500$ m) and decrease with slightly higher rate and less variability for altitudes above 1 km. However, such differences, especially at higher altitudes, are to some extent expected, since the sensitivity of the MAX-DOAS decreases rapidly with altitude for the species that are measured in the UV (Fig. 6). This means that concentrations at high altitudes are strongly constrained by the a priori profile in the retrievals of MMF. Also, parameterized algorithms (such as MAPA) have the tendency of becoming unstable when the sensitivity is low (Frieß et al., 2019).

For aerosols, the largest differences in the vertical profiles of MMF and MAPA are found in the VIS range and especially in summer and autumn. MMF yields more structured aerosol extinction profiles for altitudes between 1 and 2 km, while MAPA reports smoother, exponentially decreasing profiles. Such differences are not found in the UV retrievals during summer and autumn. It should be noted that larger discrepancies among different inversion algorithms for the species retrieved in the VIS compared to the UV have also been reported in other studies (e.g., Frieß et al., 2019; Tirpitz et al., 2021). At higher altitudes, both algorithms report greater aerosol concentrations in summer than the winter, both in the UV and VIS. Similar results were found for Thessaloniki by Siomos et al. (2018) using seasonal mean vertical profiles measured by a lidar system. In contrast, near the surface, aerosol concentrations are highest in winter and lowest in summer. This pattern can be mainly attributed to the shallower PBL in Thessaloniki during winter and autumn that shrinks to $\sim 1$ km (Georgoulias et al., 2009; Siomos et al., 2018) due to the weaker solar radiation and lower air temperatures. In the UV during winter, MAPA retrieves larger aerosol concentrations close to the ground which decrease more rapidly with altitude than MMF. However, as already discussed for HCHO, the sensitivity of the MAX-DOAS in the UV is very limited at higher altitudes and the profiles of MMF are driven towards the a priori profile. Another contributing factor for the differences in the aerosol extinction profiles between the two algorithms might be the variable $O_4$ scaling factor that is used in MAPA retrievals, while no scaling factor is applied in MMF. The effect of the $O_4$ scaling factor on the AOD is presented in the Appendix A.

## 4 Validation

In this section we present the validation results of the products retrieved by the MAX-DOAS profile analysis against ancillary data measured by other reference co-located instruments. Vertical profiles of the aerosol extinction measured by a co-located lidar system are used to validate the aerosol vertical profiles retrieved by the MAX-DOAS, while the AODs in the UV and VIS range are compared with those measured by a sun-photometer and a spectrophotometer. The $NO_2$ near-surface concentrations are compared with in situ surface measurements, but since no other sources of HCHO data are available, the MAX-DOAS derived vertical profiles, columns or surface concentrations cannot be validated.

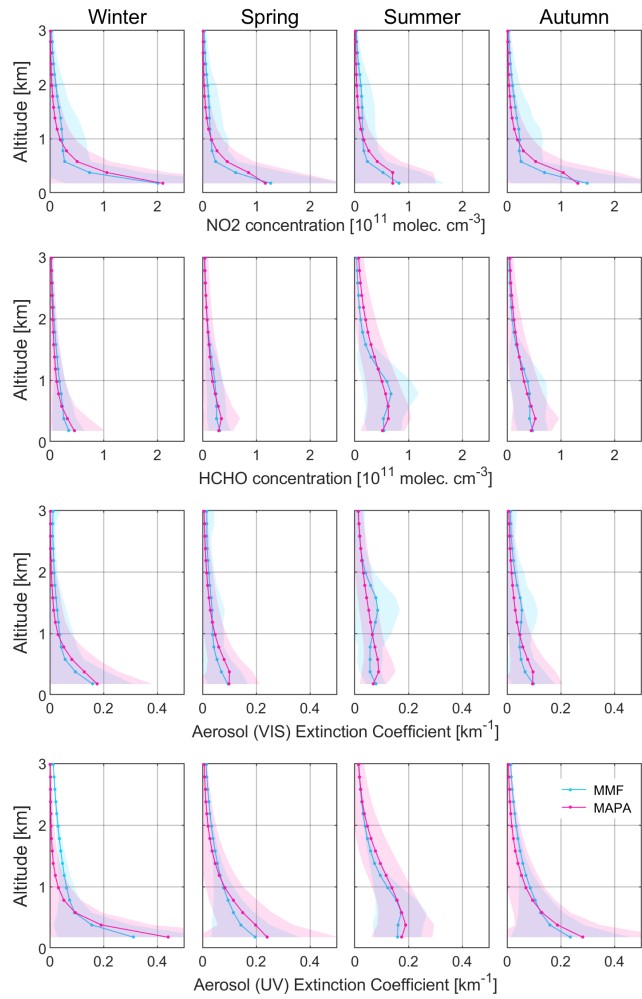

**Figure 9.** The seasonal mean vertical profiles of NO$_2$, HCHO and aerosol extinction in the VIS and UV retrieved by MMF and MAPA. Each row (1 – 4) represents the vertical profiles of different species along the four seasons (columns 1 – 4). The shaded areas represent the standard deviation around the mean for each layer and they illustrate the seasonal variability of the profiles.

## 4.1 Aerosol extinction profiles

The AOD values at 477 and 360 nm retrieved by the MAX-DOAS are compared with the AOD measured by the co-located CIMEL sun-photometer and the Brewer spectrophotometer. Quasi-simultaneous (within ± 15 minutes) measurements were found and the AODs at 477 and 360 nm were calculated using the Ångström exponent between 380 and 500 nm and the AOD at these wavelengths derived by the CIMEL. Since the Brewer's wavelength range spans up to 365 nm, climatological monthly mean Ångström exponent values, calculated from the CIMEL data, have been used to extrapolate the AOD to 477 nm. Figure 10 shows the time series of all AOD data at 477 and 360 nm (not just the quasi-simultaneous) retrieved by the three systems.

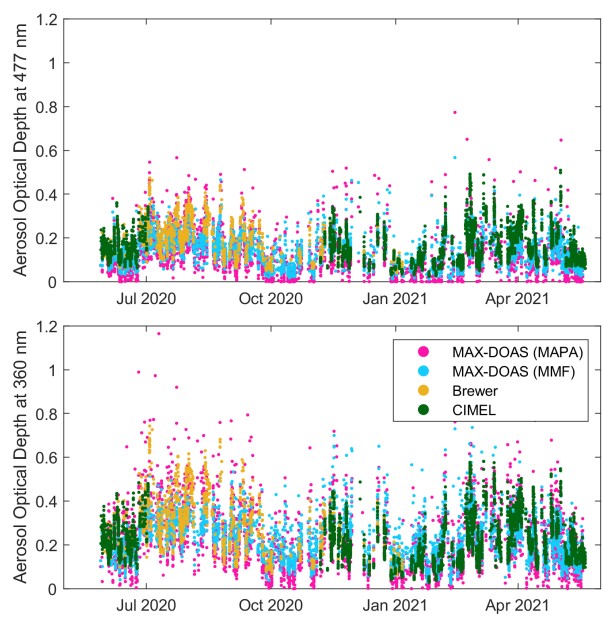

**Figure 10.** Time series of all available AOD data at 477 and 360 nm retrieved by the MAX-DOAS system, the Brewer spectrophotometer and the CIMEL sun-photometer.

**Table 4.** The flagging schemes that are applied to the retrieved products.

| Flagging scheme | Description |
| --- | --- |
| #1 | Data are flagged as valid by the flagging algorithm of MMF |
| #2 | Same as scheme #1, but for the flagging algorithm of MAPA |
| #3 | Data that are flagged as warning by either MMF or MAPA are also considered valid |
| #4 | Data are flagged as valid by both MMF and MAPA |

The CIMEL sun-photometer was not operating for approximately 4 months during the summer of 2020 due to a delay in its scheduled annual maintenance and calibration. AOD data derived by the Brewer are available until January 2021.

Since MMF and MAPA rely on their own individual flagging schemes in order to ensure that the retrieved products are of high quality, we investigate the effect of applying different flagging schemes to the data, which are listed in Table 4.

Schemes #1 and #2 correspond to the default own flagging algorithms of MMF and MAPA, scheme #4 is expected to provide 515 data of maximum quality since data are designated as valid by both algorithms, while scheme #3 rejects the error-flagged data but treats the warnings raised by MMF or MAPA as valid data. Figure 11 shows the comparison between the common AOD data derived by CIMEL, Brewer and the MAX-DOAS at 360 and 477 nm. Each column of the figure corresponds to a different flagging scheme as described in Table 4. Figure 12 presents graphically the statistics of the linear regressions (i.e., slope, offset,

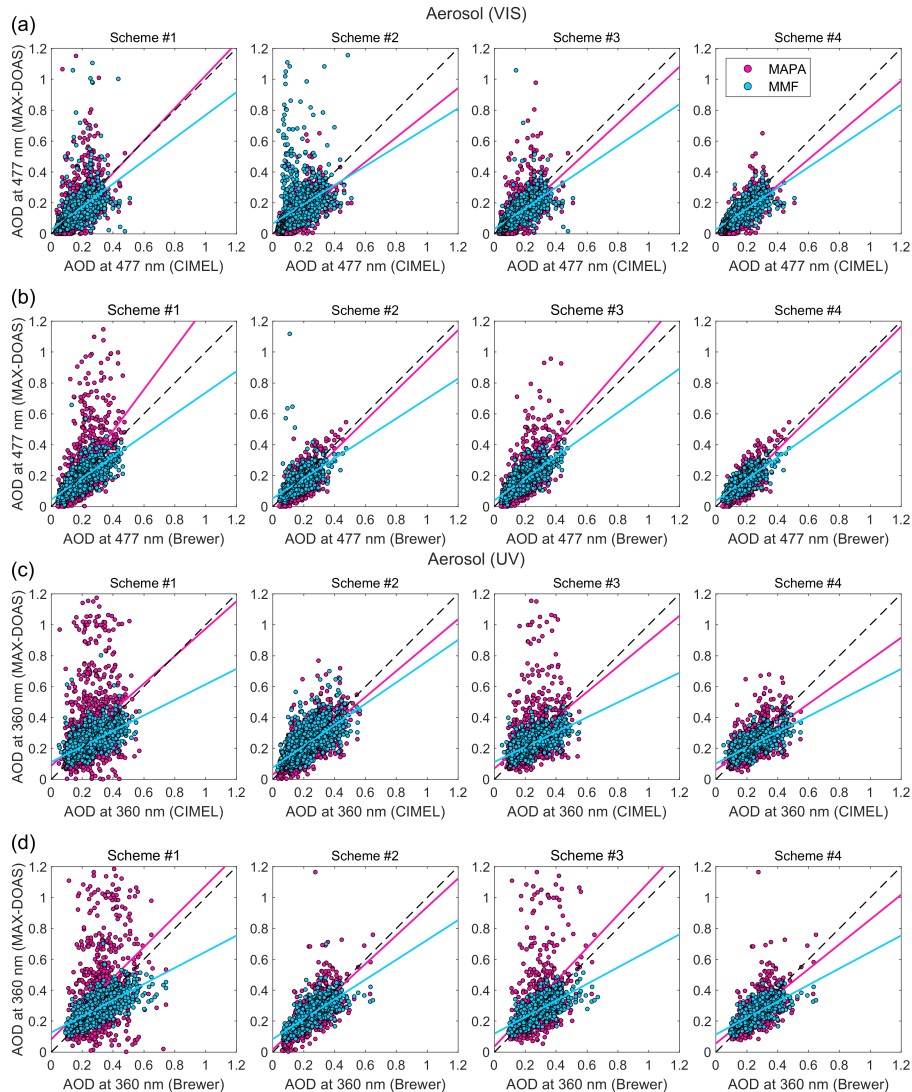

**Figure 11.** Scatter plots of the AOD retrieved by the MAX-DOAS data analyzed by MMF and MAPA against the CIMEL (**a** at 477 nm, **c** at 360 nm) and the Brewer (**b** at 477 nm, **d** at 360 nm). Each column represents data that are flagged as valid according to the flagging schemes of Table 4.

number of points and Pearson's correlation coefficient) between the reference instruments and the MAX-DOAS. The panels a – d correspond to different flagging schemes, as Figure 11.

The comparison results of the MAX-DOAS against the CIMEL are slightly different than those with the Brewer. This can probably be explained by the fact that only few collocated measurements are available and in different periods for the two reference instruments (Figure 10). In the case of the AOD at 477 nm most of the outliers are filtered-out when the flagging scheme #4 is applied and the best agreement is observed between the reference instruments and the MAX-DOAS, both for

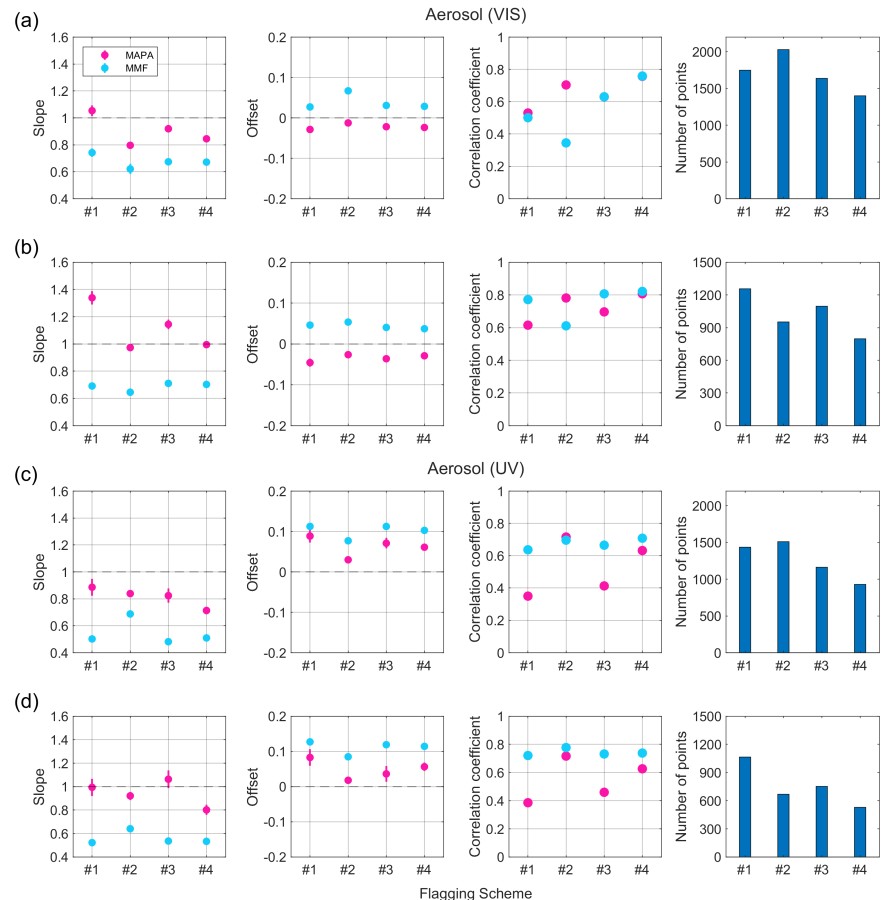

**Figure 12.** Graphical representation of the linear regression parameters (slope, offset, correlation coefficient and number of data) of the comparison between the AOD derived from MAPA and MMF against the AOD from the CIMEL (**a**, **c**) and the Brewer (**b**, **d**) at 477 nm (**a**, **b**) and 360 nm (**c**, **d**) for each flagging scheme (#1 to #4).

MMF and MAPA, with similar correlation coefficients (0.79 for the CIMEL and 0.81 for the Brewer). Compared to the CIMEL, MAPA seems to perform slightly better than MMF when each algorithm considers its own flagging, with correlation coefficients of 0.70 and 0.50, respectively (MAPA for scheme #2 and MMF for scheme #1). However, compared to the Brewer, both algorithms show very similar correlation coefficients (0.78 and 0.77, respectively). Swapping the flags between MMF and MAPA leads to worse agreement and more outliers. The results of scheme #3 indicate that some of the warning-flagged data are of lower quality and should be treated with caution.

In the case of the AOD at 360 nm, the effect of the flagging schemes is different. Here, most of the outliers are eliminated and the best overall agreement is achieved for scheme #2 (i.e., when MAPA's individual flagging algorithm is applied). This behavior is observed both for MMF and MAPA, indicating that the flagging algorithm of MAPA performs better than that of MMF in the UV. The correlation coefficients with the CIMEL data are 0.72 for MAPA and 0.70 for MMF, and with the Brewer

data are 0.72 for MAPA and 0.78 for MMF. The flagging scheme #4 removes even more data (as expected), but does not improve the comparisons. The effect of the warning-flagged data (scheme #3) is more apparent in the case of the UV and the results suggest that they should not be considered valid. The AOD derived from the MAX-DOAS, both in the UV and the VIS range, is, generally, underestimated compared to the AOD measured by the CIMEL and the Brewer. This finding is consistent with other studies (e.g., Clémer et al., 2010; Bösch et al., 2018; Davis et al., 2020; Tirpitz et al., 2021). However, it should

be noted that the AODs derived by the MAX-DOAS and the reference instruments may not always refer to the same physical quantity. The CIMEL and the Brewer use direct-sun observations to retrieve the total column amount of the aerosol extinction, while the MAX-DOAS sensitivity decreases rapidly with altitude (Sect. 3.2) and the derived AOD corresponds mainly to the lowermost tropospheric aerosol (partial AOD). Additionally, the vertical profiles retrieved by OEM-based algorithms are biased towards the a priori profile at higher altitudes (Figure 6) leading to deviations from the true profile, meaning that aerosol layers

above $\approx$ 2 km cannot be reliably retrieved (Frieß et al., 2016, 2019). Discrepancies in the AODs between the instruments are expected, usually when aerosols are present at altitudes greater than 2 km, contributing to the total AOD, but not detected by the MAX-DOAS.

During the whole period of this study only a few lidar measurements of the aerosol extinction profile are available, so the true state of the aerosol profile is generally not known. Additionally, synchronous measurements between the lidar and the

550 MAX-DOAS are even fewer, so an in-depth validation of the aerosol profiles retrieved by the MAX-DOAS needs further investigation. In this section we present the comparison of four profiles retrieved by the two systems within $\pm$ 30 min (using the MAX-DOAS profile that is closest in time with the lidar measurement), and which are indicative for the period of study. An important issue that arises in the validation of the MAX-DOAS vertical profiles is that usually the validator (in this case the lidar system) allows the detection of aerosol layers in a much higher vertical resolution than the MAX-DOAS. When the

555 true aerosol profile state is actually known and in order to compare meaningful results, the lidar profiles need to be smoothed (i.e., degraded to the sensitivity of the MAX-DOAS). Only the OEM-based algorithm provides information that can be used to smooth the lidar profiles. The information about the sensitivity is quantified by the averaging kernel according to Rodgers and Connor (2003):

$$\boldsymbol{x}_s = \boldsymbol{x}_a + \mathbf{A}\left(\boldsymbol{x} - \boldsymbol{x}_a\right) \tag{2}$$

where $\boldsymbol{x}_s$ is the smoothed lidar profile, $\boldsymbol{x}_a$ and $\mathbf{A}$ are the a priori profile and the averaging kernel of the OEM-based retrieval and $\boldsymbol{x}$ is the initial lidar profile. Deviations of the smoothed lidar profile at each altitude depend on the a priori profile and the sensitivity of the MAX-DOAS at this altitude (Davis et al., 2020). Since the sensitivity of the MAX-DOAS decreases with altitude, the application of the averaging kernels is expected to smooth the true profiles towards lower altitudes. However, since MAPA does not quantify the sensitivity, a similar smoothing cannot be performed for MAPA retrievals and thus, the aerosol

extinction profiles are directly compared with the initial lidar profiles. Another point that should be noted is the differences in the operational principles of the two instruments. The lidar retrieves the vertical profile from the air mass that is located overhead, while the MAX-DOAS scans through different air masses along the line of sight of the telescope during an elevation

sequence (Gratsea et al., 2021). Its effective horizontal distance is of the order of a few kilometers and increases at elevation angles close to the horizon. Thus, differences in the retrieved extinction profiles are expected, especially at locations with large horizontal inhomogeneities of aerosols. As already mentioned (Sect. 2.7.3) a constant climatological Lidar Ratio of 50 sr was assumed for the channel of 532 nm and was applied to the backscatter profiles in order to retrieve the extinction, which may also result in uncertainties of the validator's product. So, in this study the comparisons are focused on the shape of the profiles and the retrieved aerosol layer heights rather than on the absolute values of the aerosol extinction.

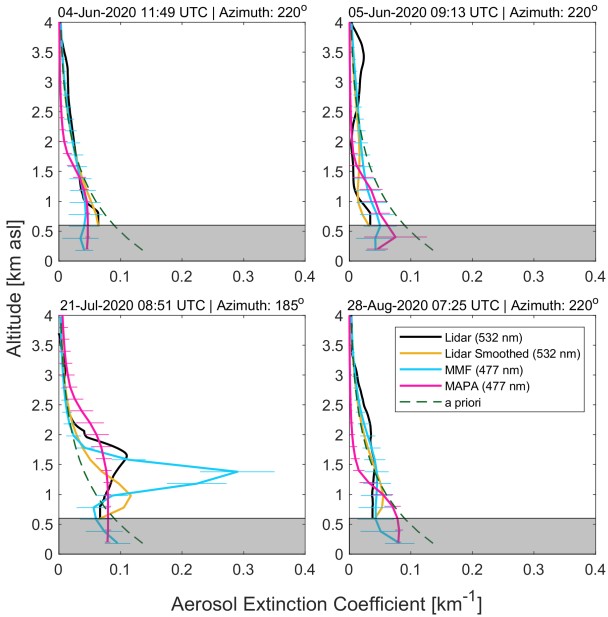

**Figure 13.** Four cases of aerosol vertical profiles measured by the MAX-DOAS (cyan for MMF and magenta for MAPA) and the lidar (black for the original and yellow for the smoothed profiles). The shaded area represents altitudes, where the lidar is not capable of retrieving the aerosol extinction profile accurately due to the overlap effect.

Figure 13 presents the comparison of four aerosol extinction profiles in the VIS retrieved by the MAX-DOAS and the lidar. The lidar profile is trustworthy only above a certain altitude (approximately 0.6 km) owing to the geometry of the telescope and the emitted laser beam, which prevents a fraction of the backscattered radiation to reach the detector at altitudes close to the surface (overlap effect). Thus, the aerosol extinction retrieved by the lidar below 0.6 km is not presented. Since the MAX-DOAS profile retrievals in the UV are sensitive at altitudes closer to the ground (see discussion in Sect. 3.2), where the lidar system is not, the profiles for 360 nm are excluded from the analysis. Less information content obtained for species measured in the UV was also apparent in other studies (e.g., Schreier et al., 2021; Tirpitz et al., 2021). In Figure 13 the yellow line corresponds to the smoothed lidar profile, which is degraded to the sensitivity of the MAX-DOAS according to Eq. 2. In general, the aerosol vertical profiles retrieved by the MAX-DOAS can realistically estimate the shape of the true state, even though some differences appear between MMF and MAPA. The agreement between the shape of the MMF profiles and

the smoothed lidar profiles is much better than for MAPA. This is expected since the initial lidar profile is degraded to the sensitivity of the MAX-DOAS using the averaging kernels derived by MMF and the a priori profile of the retrieval. On 4 and 5 June the aerosol load is low and both algorithms report similar profiles. In both cases the MAX-DOAS profiles fit successfully the shape of the true profile. However, on 5 June the shape of the profiles is similar only up to $\approx 2$ km. Aerosol layers between 2 and 4 km that are detected by the lidar cannot be well retrieved by the MAX-DOAS, due to its very limited sensitivity at these altitudes (Figure 6). In the other two cases MMF and MAPA provide different profiles. Discrepancies in the aerosol extinction profiles retrieved in the VIS by MMF and MAPA are also found in the seasonal mean vertical profiles (Sect. 3.5), especially during summer. On 21 July MMF reports a more structured aerosol profile with a distinct aerosol layer at about 1.3 km, while MAPA reports a smoother profile with a thick layer spanning from the surface up to $\approx 2$ km. On 28 August, even though MMF and MAPA report profiles of different shapes below $\approx 1$ km, the profiles agree reasonably well with the lidar profile. At higher altitudes MMF is biased towards the a priori profile due to the limited sensitivity, while the aerosol extinction retrieved by MAPA decreases rapidly. Despite the observed differences, the results of the comparisons are promising, indicating that the analysis of the MAX-DOAS data can provide a generally good estimation of the vertical aerosol extinction profiles over Thessaloniki. However, further investigation is required in order to assess the differences in the aerosol profiles provided by the two systems, but also between the two inversion algorithms, when more collocated measurements become available. Such studies will be further facilitated with a new lidar system with improved overlap height that is currently under development, which will allow the retrieval of the aerosol profiles at altitudes closer to the ground, where the MAX-DOAS shows higher sensitivity.

## 4.2   NO$_2$ surface concentration

In Figure 14 we present a comparison of near-surface concentrations derived from the MAX-DOAS data (i.e., the average concentration 200 m) with in situ NO$_2$ measurements. The small dots represent the hourly mean values, while the solid lines refer to the daily mean concentrations. The comparison is only performed for the hourly mean concentrations derived by the two systems, while the daily mean concentrations are shown only for a qualitative comparison. The MAX-DOAS hourly mean values are horizontally averaged from all azimuth viewing directions. A dataset from June 2020 to March 2021 (about 10 months) is considered in this study, in which both MAX-DOAS and in situ measurements were available. The MAX-DOAS reports systematically lower NO$_2$ concentrations than the in situ by $\sim 55 - 60\%$. Since the concentrations retrieved by the MAX-DOAS are averaged along a horizontal path of a few km, which may extend over the bay, whereas the in situ data refer to a specific location (point measurements), the MAX-DOAS is generally expected to report lower values from the air-quality station, which is also occasionally affected by local emissions. Differences in the retrieved concentrations may also arise due to the slightly different altitudes of the measurement sites. Similar results have been found in other studies (e.g., Friedrich et al., 2019; Chan et al., 2020; Dimitropoulou et al., 2020). Both MMF and MAPA perform well for the retrieval of the NO$_2$ near-surface concentrations. Even though the in situ site is located opposite to the MAX-DOAS system's azimuth viewing directions (see Figure 2), the correlation coefficients of both algorithms are still high, suggesting that strong NO$_2$ horizontal inhomogeneities are less likely to occur in Thessaloniki, at least during the period of study. The effect of the different flagging

schemes is not as strong as for aerosols, except for MAPA when MMF's flagging (scheme #1) is applied to the data. The performance of MMF is slightly better than MAPA's with fewer outliers and higher correlation coefficients, even though it reports as valid a much larger fraction of data (Table 3). The results of flagging scheme #3 indicate that the warning-flagged data could also be considered valid. This could be explained by the fact that a large part of the flagged data is related to the effects of clouds. As shown in previous studies (e.g., Wang et al., 2017b), under most cloud conditions (except for fog and optically thick clouds), the trace gas vertical profiles, columns and near-surface concentrations can still be retrieved, while the aerosol retrievals are stronger affected.

## 5   Summary and conclusions

In this study we have retrieved vertical profiles of aerosols, $NO_2$ and HCHO for the first time in Thessaloniki, Greece using MAX-DOAS observations by applying an OEM-based inversion algorithm (MMF) and a parameterized algorithm (MAPA). Their performance is evaluated by intercomparing the dSCDs simulated by the forward models, the integrated columns (i.e., VCDs for trace gases and AODs for aerosols), the trace gas near-surface concentrations and the seasonal mean vertical profiles derived by the two algorithms. The products that are retrieved by the inversion analysis of MAX-DOAS measurements using MMF and MAPA are compared with ancillary data measured by other reference instruments. This study provides the basis for future research activities, e.g., the investigation of the spatio-temporal variability of trace gas and aerosol profiles over Thessaloniki.

The tropospheric column densities of $NO_2$ are in excellent agreement (slope very close to unity and $R = 0.982$), while for HCHO, even though a generally good correlation is found ($R = 0.927$), deviations from unity in the slopes are observed, which can be attributed to discrepancies between the HCHO dSCDs simulated by the forward models of MMF and MAPA and the limited sensitivity of the MAX-DOAS in the UV, especially at higher altitudes. Concerning aerosols, a better agreement between MMF and MAPA is found for the AOD at 477 nm than at 360 nm due to the increased SNR in the VIS range and the stronger effect of the $O_4$ scaling factor on the retrieved AODs in the UV. No clear azimuth dependence is observed for any of the retrieved species. The seasonal mean vertical profiles retrieved by MMF and MAPA are generally in good agreement. The largest discrepancies are found for the aerosol extinction profiles in the VIS and especially during summer.

The AODs retrieved by the MAX-DOAS are validated by comparison with measurements of a CIMEL sun-photometer and a Brewer spectrophotometer. Four flagging schemes were applied to the MAX-DOAS derived data and their effect on different products is evaluated. However, no robust conclusion could be drawn about which flagging algorithm shows an overall better performance. A generally good qualitative agreement is found for both VIS and UV wavelengths (with correlation coefficients up to 0.8). The negative bias that is observed from the reference instruments is probably mostly due to the limited sensitivity of the MAX-DOAS in retrieving aerosol information at higher altitudes, especially in the UV. The results also indicate that using an intersected dataset derived by applying both flagging algorithms to the data, improves the agreement of AODs at 477 nm, however, a similarly good agreement is not observed in the UV, where the flagging algorithm of MAPA performs better. Four cases of aerosol extinction vertical profiles at 477 nm are compared with profiles measured by a co-located lidar

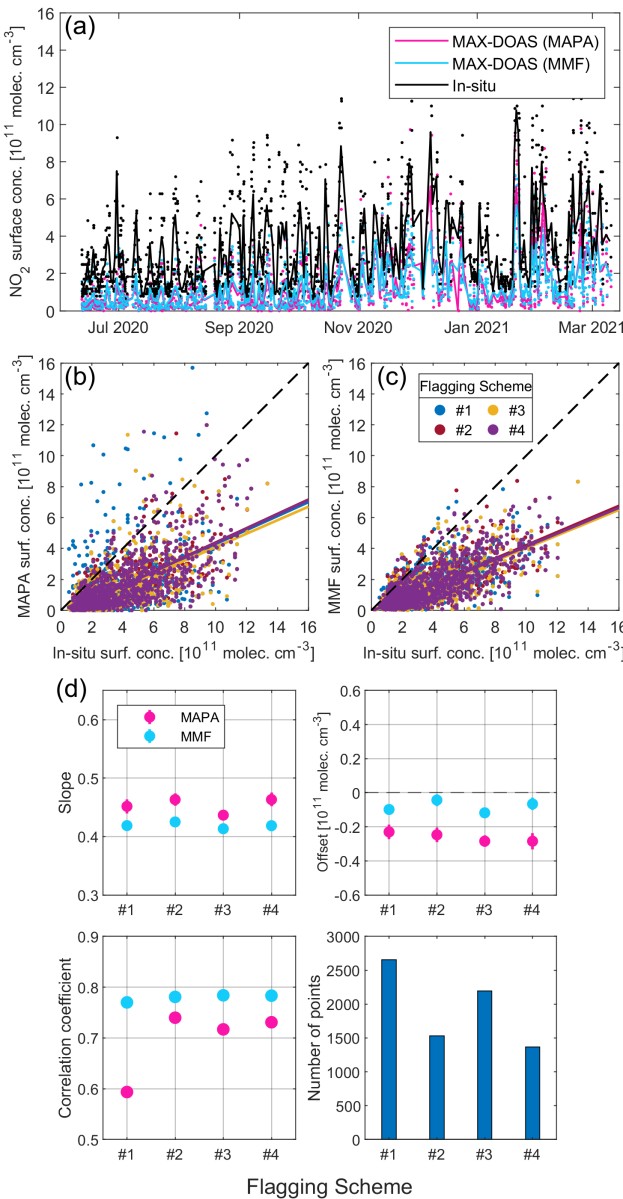

**Figure 14. (a)** Time series of hourly mean NO₂ surface concentrations derived from MAX-DOAS by MMF (cyan) and MAPA (magenta) and from in situ measurements (black). The solid lines correspond to the time series of daily means. Panels **b**, **c** are the corresponding scatter plots colored by the four flagging schemes that are applied to the MAX-DOAS data and panel **d** shows the statistics of the comparisons.

system. The MAX-DOAS was found to provide a generally good estimation of the shape of the profile. The $NO_2$ near-surface concentrations are compared with in situ $NO_2$ observations, where the effect of the different flagging schemes is not found to have such a strong impact as for aerosols. The concentrations from both MMF and MAPA are in good agreement with the in situ measurements in terms of variability, but are highly biased by approximately 60%. MMF shows a slightly better performance ($R = 0.78$) compared to MAPA ($R = 0.73$).

The effect of the $O_4$ scaling factor is also investigated by comparing the integrated columns of MMF and MAPA and also by comparing the AODs derived by MAPA for different values of the scaling factor with AODs measured by the CIMEL and the Brewer. The effect of the $O_4$ scaling factor has a stronger impact on aerosols than on trace gases (where the effect is minor). The fixed value of 0.8 for the scaling factor, which is supported by many studies, does not seem to be suitable for the measurements at Thessaloniki.

## Appendix A: Effect of the $O_4$ scaling factor

The $O_4$ scaling factor (SF) was introduced by Wagner et al. (2009) in order to remove the systematic discrepancies appearing between measured and simulated $O_4$ dSCDs. Uncertainties of the $O_4$ cross sections and/or its temperature and pressure dependence, aerosol optical properties and RTM errors have been suggested as possible causes for these discrepancies. Several studies have confirmed the idea of the $O_4$ scaling factor and have shown that applying a SF (commonly using a value between 0.75 and 0.9) is indeed necessary (Wagner et al., 2009; Clémer et al., 2010; Irie et al., 2011; Wang et al., 2014; Vlemmix et al., 2015b; Wang et al., 2016; Frieß et al., 2016; Wagner et al., 2019, 2021). However, other studies have not supported this requirement (Spinei et al., 2015; Ortega et al., 2016; Seyler et al., 2017; Wang et al., 2017a, b). Although the need for an $O_4$ scaling factor for retrieving aerosol information from MAX-DOAS measurements has been extensively discussed (Wagner et al., 2019), its physical mechanism is not understood and still remains an unresolved issue.

Since MAPA provides the option of scaling the modeled $O_4$ dSCDs, we have investigated the effect of the SF on the comparisons between the products of the two profiling algorithms and between AOD derived by MAPA and the reference instruments. We selected three fixed values (i.e., 1, 0.9 and 0.8, referred hereafter as SF1.0, SF0.9 and SF0.8) and a variable SF (SFvar). Figure A1 shows the effect of the $O_4$ SF on the comparison of trace gas VCDs and AOD derived by MAPA and MMF (same as Figure 7 without accounting for the different azimuth directions). Like in Figure 7, the regression results are based on an ODR and also the retrievals of aerosols in the UV are based on the flagging algorithm of MAPA. Since MMF does not take into account a scaling for $O_4$, the use of different SFs in MAPA leads to substantial differences in the regression slopes and correlation coefficients when comparing the AODs of the two algorithms, both for 360 and 477 nm. The closest to unity slopes and the highest correlation coefficients are found, as expected, when no scaling factor is applied (SF1.0) for both wavelengths. The slopes of the fitting are 1.18 and 1.07 for the AODs at 477 and 360 nm, respectively, while the correlation coefficients are $\sim 0.89$ and $\sim 0.88$. Especially for the AOD in the UV, the agreement between MMF and MAPA for the variable SF substantially declines and the scatter increases. The worst results appear for SF0.8 leading also to substantial reduction in the reported valid data. The use of a scaling factor doesn't seem to affect the retrieved VCDs for $NO_2$ and HCHO, at least as

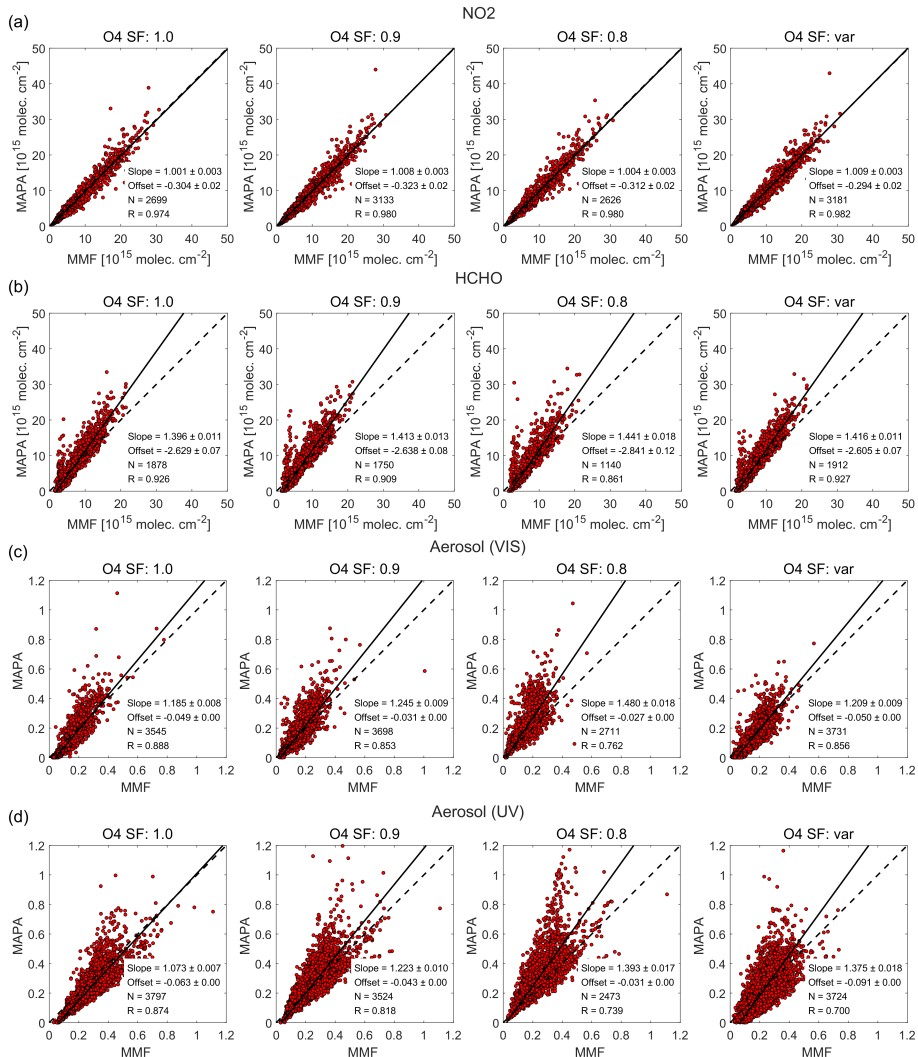

**Figure A1.** Scatter plots of the integrated columns of $NO_2$, HCHO, AOD at 477 and 360 nm (**a** – **d**, respectively) retrieved by MMF and MAPA for various $O_4$ scaling factors (columns 1 – 4).

it concerns the slope and correlation coefficient of the regression, but there is some effect on the number of the data reported as valid, especially for SF0.8. Opposite to aerosols, the best correlation of the retrieved $NO_2$ and HCHO columns is achieved when using the SFvar instead of the SF1.0.

Figure A2 presents the comparison of the AODs at 360 and 477 nm retrieved by the MAX-DOAS (using MAPA) with the AODs calculated by the CIMEL and the Brewer for different $O_4$ SFs. For consistency, the individual flagging algorithm of MAPA is used for the retrievals both in the UV and VIS. The differences in the slopes and correlation coefficients among the different SFs are rather small, with the former dominated mainly by the noise in the measurements, as discussed in Sect. 4.

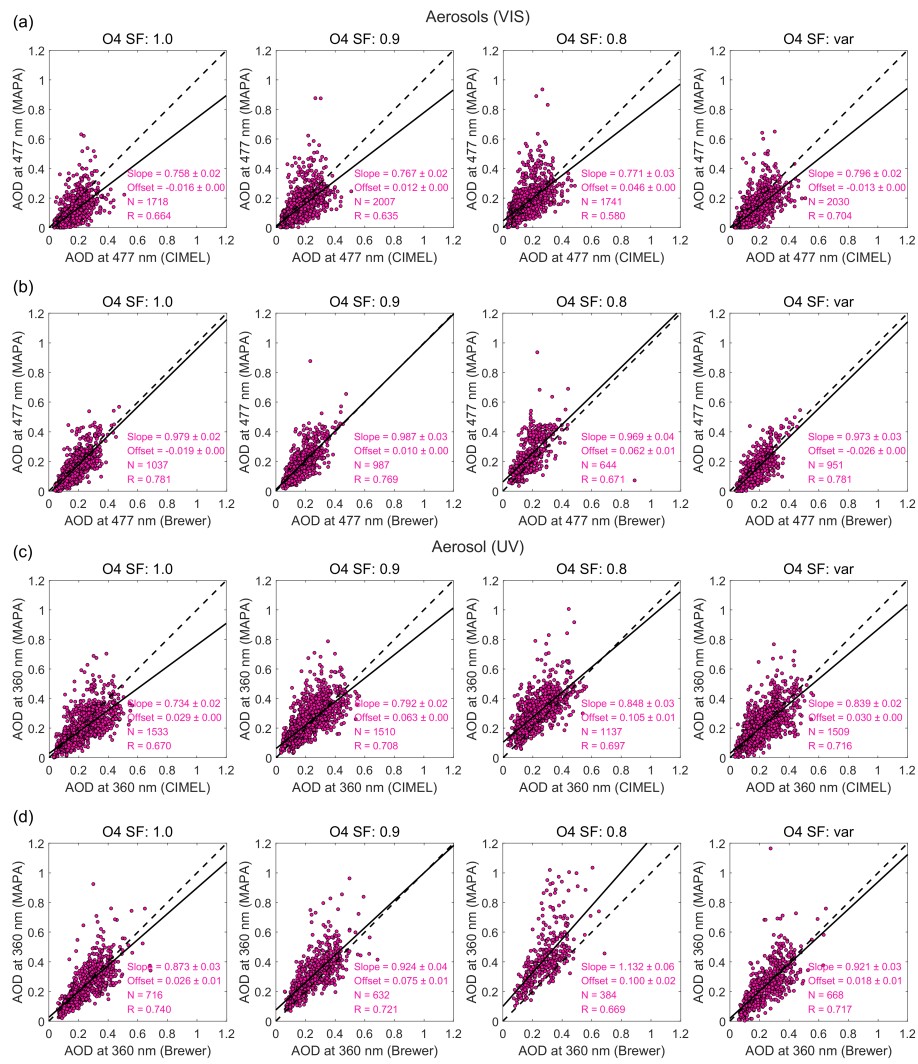

**Figure A2.** Comparison of the AODs at 477 nm (**a**, **b**) and 360 nm (**c**, **d**) derived by MAPA with the AODs measured by the CIMEL (**a**, **c**) and the Brewer (**b**, **d**) for different $O_4$ SFs (columns 1 – 4).

The results of SF1.0 (i.e., no scaling factor) show a similar performance with the SFvar indicating that the most suitable $O_4$ SF value for Thessaloniki would be closer to unity. As for the AOD, the SF0.8 flags as invalid a larger fraction of the data compared to the other SFs in all cases and the agreement between the MAX-DOAS and the reference instruments gets worse with decreased correlation coefficients and larger offsets and slopes. Hence, the SF0.8 that is supported by many studies for
achieving better agreement between the MAX-DOAS and sun-photometers (Wagner et al., 2009) is found to be too small for the profiling of the MAX-DOAS measurements at Thessaloniki.

This is also supported by the frequency distribution of the fitted $O_4$ SFs in the UV and VIS ranges, shown in Figure A3a. The median $O_4$ SF fitted by MAPA is $0.964 \pm 0.125$ and $0.964 \pm 0.115$ at 477 and 360 nm, respectively. The histograms indicate that for most elevation sequences a scaling factor close to unity is required to bring measured and simulated $O_4$ dSCDs into agreement. The SF0.8 seems to be too low for the retrievals in Thessaloniki for both spectral ranges. However, it should be noted that an apparent seasonal pattern in the fitted $O_4$ SF is observed both at 477 and 360 nm, shown in Figure A3b. In order to remove any possible effects of the seasonal variability of the SZA, only the $O_4$ SFs for which $65^o < SZA < 75^o$ are presented. The maximum $O_4$ SFs values are reported in August ($\sim 1.02$) at 477 nm and in September ($\sim 1.05$) at 360 nm, while the minimum $O_4$ SF values are found in February – March for both wavelengths (i.e., $\sim 0.86$ at 477 nm and $\sim 0.91$ at 360 nm). This seasonal variability could partly be explained by the temperature dependence of the $O_4$ absorption. Since the absorption is stronger at lower temperatures, higher $O_4$ SFs are generally expected during summer than in winter. The seasonal pattern could also be related to the similar seasonal variability of the AOD. In general, higher AODs are observed over Thessaloniki in summer than in winter (Kazadzis et al., 2007; Giannakaki et al., 2010; Siomos et al., 2018; Fountoulakis et al., 2019). The $O_4$ SFs for the two wavelengths show a similar, but not identical seasonality and thus, further investigation is required when more MAX-DOAS data become available.

**Appendix B: Comparison with the VCDs calculated using the geometrical approximation**

In this section, we compare the trace gas VCDs obtained from the integration of the vertical profiles retrieved by MMF and MAPA with those derived by other methods used in the past at Thessaloniki, in order to establish a link with the VCDs reported in former studies (e.g., Drosoglou et al., 2017; Skoulidou et al., 2021). As already discussed in Sect. 2.3 the main product that is derived from DOAS spectral analysis is the trace gas dSCD at different viewing directions. However, the conversion of dSCDs to VCDs is usually challenging. The easiest approach that has been adopted for several years is the so-called geometrical approximation (Hönninger and Platt, 2002). This methodology considers only the geometric light path through the troposphere for the attenuation of radiance at an elevation angle $\alpha$ and thus the AMF (Solomon et al., 1987) that is required for the calculation of the VCD can be geometrically approximated (Hönninger et al., 2004) by:

$$\mathrm{AMF}_\alpha = \frac{1}{\sin(\alpha)} \tag{B1}$$

This approach has been proven potentially appropriate, when higher elevation angles are used (typically 30º and/or 15º) under low aerosol conditions (Wagner et al., 2010). However, for a more accurate calculation of the true AMF, several other parameters must be taken into account, such as the solar position, viewing geometry, ground albedo, wavelength and aerosol properties (Hönninger et al., 2004). High AOD values are not infrequent in Thessaloniki, especially during summer (e.g., Kazadzis et al., 2007; Fountoulakis et al., 2019). Thus, for a more accurate calculation of the VCD, Drosoglou et al. (2017) calculated dAMF LUTs based on RTM simulations using several parameters (such as the SZA, the AOD, the elevation angle and the azimuth angle relative to the solar azimuth), appropriate for the fitting windows of the spectrometers used in that study. Since in the current study a different spectral range was used (Sect. 2.2), these LUTs could not be used here.

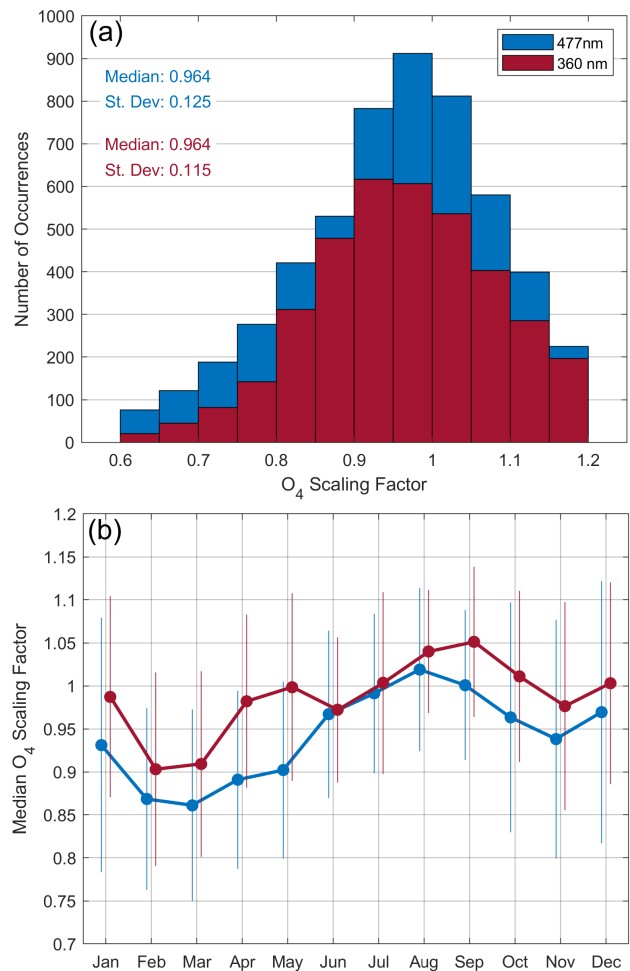

**Figure A3.** Frequency distribution of the fitted $O_4$ scaling factor at 477 nm and 360 nm **(a)** and its seasonal variability **(b)**. The error bars represent the standard deviation around the monthly averages.

Tropospheric dSCDs measured at elevation angles of 30 and 15°, relative to the zenith, are converted to VCDs by applying the geometrical approximation and their average is calculated and used when they agree to at least within 50%. This filtering is necessary since these dSCDs (especially at 30°) are much smaller than those measured at lower elevation angles and are associated with larger fitting errors (especially for HCHO). For the VCDs derived from the profiles the default flagging scheme was applied, i.e., when they are flagged as valid by both MMF and MAPA (scheme #4 of Table 4). In this comparison, no discrimination of the sky conditions (aerosol and cloud) is made.

The comparison for $NO_2$ and HCHO is shown in Figure B1. The number of available data for HCHO is much lower than for $NO_2$ due to the fewer valid profiles retrieved by the inversion algorithms (Table 3) and also because HCHO dSCDs derived from this MAX-DOAS system contain more noise, due to weak SNR in the UV, leading to larger differences between the VCDs

obtained from 30 and 15º elevation angles. For $NO_2$, the profile-derived VCDs compare well to the VCDs obtained from the geometrical approximation with high correlation coefficients ($R = 0.96$ and $R = 0.93$ for MMF and MAPA, respectively). For

HCHO the correlation coefficients are lower ($R = 0.93$ for MMF and $R = 0.89$ for MAPA). The discrepancies in the results between MMF and MAPA are consistent with the results of Figure 7 and are discussed in Sect. 3.3. $NO_2$ is typically located at lower altitudes than HCHO, mainly because it is produced by emission sources close to the surface (see discussion in Sect. 3.5). Other studies have shown that the error of the geometric VCDs is usually less than 20% compared to the integrated vertical profiles for trace gases that are located below 1 km, while for trace gases that are located at higher altitudes the

geometrical approximation is less accurate since the effect of aerosols becomes more important (e.g., Shaiganfar et al., 2011; Wagner et al., 2011). This finding is consistent with the results of Figure B1, where lower correlation coefficients are found for HCHO as HCHO layers can be vertically extended to higher altitudes, making the geometrical approximation less appropriate. Other studies (e.g., Kumar et al., 2020) have also shown that aerosols and clouds further limit the accuracy of the geometrical approximation, yet their effect is not investigated in this study. However, in both cases, the VCDs that are calculated with the

geometrical approach can be generally considered a relatively good estimation of the VCDs that are obtained by the vertical profiles analyzed in this study. The calculation of VCDs using model-derived dAMFs would possibly further improve the comparison with the profile derived VCDs. Unfortunately, the dAMFs that were used in Drosoglou et al. (2017), do not include longer wavelengths that are appropriate for the dSCDs derived by the MAX-DOAS system used in this study.

*Author contributions.* DK developed the intercomparison and validation strategy of the two inversion algorithms, analyzed the MAX-DOAS

data of Thessaloniki, performed the offline retrievals, conducted the data analysis and wrote the manuscript. MMF provided the MMF source code, supported and guided DK during the whole time for the proper use of the inversion algorithm and provided a lot of feedback for the interpretation of the OEM-based results. SB and TW provided the MAPA source code along with useful information about the retrievals and contributed to scientific discussions. KAV provided the lidar extinction profiles, IF and AK the Brewer-derived AOD and PT the in situ $NO_2$ data. MVR, FH and DB reviewed the paper. AB supervised the whole study and provided general guidance for the manuscript preparation.

All authors discussed, commented and helped reviewing the manuscript.

*Competing interests.* I declare that I or my co-authors have competing interests as follows: Steffen Beirle is associate editor of AMT. Michel Van Roozendael is associate editor of AMT. Thomas Wagner is chief-executive editor and associate editor of AMT.

*Acknowledgements.* This research is co-financed by Greece and the European Union (European Social Fund- ESF) through the Operational Programme "Human Resources Development, Education and Lifelong Learning 2014-2020" in the context of the project "Strengthening

Human Resources Research Potential via Doctorate Research – 2nd Cycle" (MIS 5000432). We acknowledge support of this work by the project "PANhellenic infrastructure for Atmospheric Composition and climatE change" (MIS 5021516) which is implemented under the Action "Reinforcement of the Research and Innovation Infrastructure", funded by the Operational Programme "Competitiveness, En-

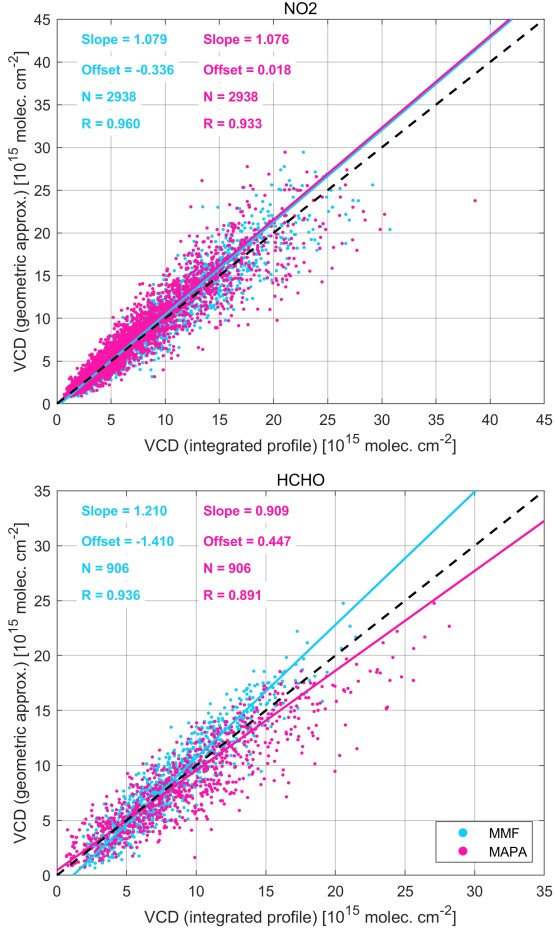

**Figure B1.** Comparison of NO$_2$ (upper) and HCHO (bottom) VCDs calculated using the geometrical approximation with the VCDs obtained from the integration of the vertical profiles retrieved by MMF (cyan) and MAPA (magenta).

trepreneurship and Innovation" (NSRF 2014-2020) and co-financed by Greece and the European Union (European Regional Development Fund). This study has been partly supported by the FRM$_4$DOAS project (ESA contract no. 4000118181/16/I-EF). We also thank Caroline Fayt (caroline.fayt@aeronomie.be) and Thomas Danckaert for the free use of the QDOAS software and Robert Spurr for providing the VLIDORT radiative transfer code package.

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
