# Peer review of "Retrieval of tropospheric aerosol, NO2 and HCHO vertical profiles from MAX-DOAS observations over Thessaloniki, Greece: Intercomparison and validation of two inversion algorithms"

_Atmospheric Measurement Techniques, 2021_

## Author Comment (AC1)

Response to Referee #1

We would like to thank the reviewer for carefully reading the manuscript and for providing helpful comments, remarks and suggestions. You can find below our responses in red after each individual comment:

Due to the reordering of Sect. 3 (Results) and adding new figures, all references to text and figures of the manuscript that are mentioned below refer to the revised version, while in square brackets [] they refer to the preprint.

e.g., Figure 6 [8] refers to Figure 6 of the revised version and to Figure 8 of the preprint.

General comments

Karagkiozidis et al. present a comprehensive comparison and validation study of two MAX-DOAS profiling algorithms. The algorithms retrieve trace gas and aerosol profiles from MAX-DOAS observations over Thessaloniki, Greece.

The manuscript is well written, the analyses have been performed thoroughly and the conclusions are interesting.

However, while reading this document, I was wondering, what is the aim of this study? From the title, I expected a characterization of the temporal and spatial distribution of NO2, HCHO and aerosols over Thessaloniki. But the authors focused mainly on the comparison and validation of two profiling algorithms. Algorithms which have already been validated in other studies! In my viewpoint, the authors should change the manuscript slightly in order to go more in the direction of either a pure algorithm validation/verification paper or a characterization paper of Thessaloniki's trace gas/aerosol distribution.

If the authors decide for case 1, I would expect a detailed comparison of vertical profiles. If validation is not possible due to sparse measurements of ancillary instruments, please add a comparison of temporal/spatial mean profiles of both algorithms. I was also wondering if MAPA retrieves concentrations in higher altitudes compared to MMF? On the other hand, MMF does not retrieve small VCD's even though the correlation with in situ data is high. Does the a priori SH of 1km leads to this constrain? If the manuscript is modified based on these suggestions, please change the title accordingly.

In case the authors decide for a characterization paper of the tropospheric composition over Thessaloniki, I would expect a discussion of weekday to weekend variations. I would also expect diurnal variation plots. Furthermore, in this case, the analysis of HCHO is insufficient. Even though validation is not possible we learn nothing about the spatial distribution of HCHO from your study. You neither show HCHO profiles nor do you talk about possible sources (for all species).

In both cases, I would love to see some contour plots of seasonal mean profiles for all species.

The point made by the reviewer about focusing on the intercomparison and validation of two profiling algorithms in this study, instead of characterizing the aerosol and trace gas distribution at Thessaloniki is very reasonable. The main motivation for conducting this study was the absence of MAX-DOAS derived information on the vertical distribution of aerosols and trace gases in the troposphere at Thessaloniki, even though MAX-DOAS measurements are regularly performed since 2014. As the reviewer states,

MMF and MAPA have been tested in other studies and based on their performance they were accepted and adopted by the FRM$_4$DOAS project. By using two well-performing algorithms, we expect to retrieve profiles (and subsequently VCDs/AODs, surface concentrations, etc.) of higher quality and we tried to validate them with ancillary, reference data when available. In addition, we wanted to evaluate the ability to retrieve vertical profiles with a recently installed MAX-DOAS system that has superior characteristics than the instruments used in the past (e.g., wavelength range, spectral resolution, signal to noise ratio, field of view). We agree that additional analysis of the data (i.e., characterizing the aerosol and trace gas distribution at Thessaloniki) would be scientifically meaningful. However, in our belief, this paper already contains a lot of information and including an additional geophysical analysis would significantly increase the size of the paper and make it difficult for the reader to follow. Such analysis could be part of another paper in the future. Thus, we have followed the first suggestion of the reviewer and focused to an intercomparison/validation study and we have revised the title of the paper to "Retrieval of tropospheric aerosol, NO$_2$ and HCHO vertical profiles from MAX-DOAS observations over Thessaloniki, Greece: Intercomparison and validation of two inversion algorithms".

The reviewer correctly pointed out that according to the title, vertical aerosol and trace gas profiles should be presented, yet they are not shown in any of the figures. As the reviewer suggests, we included an additional subsection in the results section were we discuss seasonal mean profiles for all species retrieved by the two algorithms. In this subsection the vertical profiles are intercompared, the variability of each algorithm is discussed along with possible sources for each species. Similar comments about the absence of the vertical profiles in the plots and insufficient information about HCHO were made also by reviewer #2. Along with the seasonal mean vertical profiles of HCHO we included an extra appendix showing the comparison of the VCDs retrieved from the integration of the profiles with the VCDs that are obtained using the geometrical approximation in order to make a link with previous datasets.

Please also add the following points:

1. Even though the applied flags have been applied elsewhere, please add a table of flags for each algorithm in the appendix. I guess that flagging thresholds might differ for UV and vis?

As the reviewer states, flags and thresholds of the algorithms have already been discussed in detail in other studies (e.g., Beirle et al., 2019). However, it is true that a short discussion about flagging should be present in the manuscript. Instead of adding an extra table that contains information about the flags and thresholds in the appendix, for better readability we included a short discussion for each algorithm in the relevant sections (2.4 for MMF and 2.5 for MAPA). The default flagging thresholds in MAPA are the same for the UV and VIS spectral range. The user can change the thresholds and even set different values for the UV and VIS. Yet, in this study we use MAPA with the default configuration.

2. What is the conclusion of your flagging scheme discussion? I would consider your results as unclear. Maybe there is no clear conclusion to be made?

It is true that no clear conclusion could be achieved based on this study about which flagging algorithm (MMF's or MAPA's) performs overall better (the text has been revised). However, the flagging

algorithms have been evaluated during the comparison of MAX-DOAS derived products (AODs and $NO_2$ surface concentrations) with reference instruments. For the AODs in the visible range, higher correlation coefficient values are achieved between the MAX-DOAS and the CIMEL/Brewer when both MMF and MAPA contribute to the flagging; yet, this is not the case for AODs in the UV, where MAPA's flagging algorithm leads to better agreement. For $NO_2$ surface concentrations, the effect of the flagging scheme is not as strong as for aerosols. The combined flagging leads to slightly better agreement with the in situ data. Individually, the MMF's flagging performs slightly better than MAPA's since it results in higher correlation with the in situ data even though a much smaller fraction of the data is flagged as invalid (Table 3).

3. Please add a short discussion of NO2 retrieved in the UV. You have mentioned that HCHO and aerosols in the UV might be negatively affected by increased spectral noise. Is there a similar conclusion for UV NO2?

The reason for this suggestion of the reviewer is fully understood. However, since this instrument allows for the retrieval of $NO_2$ dSCDs in the visible range (425 – 490 nm), we don't normally retrieve $NO_2$ in the UV range, where larger fitting errors are expected due to the increased spectral noise in this region. Even though $NO_2$ dSCDs can still be retrieved in the UV, a lot of computational effort is required in order to retrieve its vertical profiles for approximately 1 year of data. Besides, including an "additional" species in the analysis would increase the size the paper without providing much extra valuable information (except for the differences between the UV and VIS).

4. Please add a short discussion of possible issues of your aerosol retrieval due to the inaccurate Henyey-Greenstein phase function at the proper sections in your manuscript.

For most viewing directions, the respective choice of the aerosol phase function has only a minor effect, because the main effect of aerosols is that they reduce the visibility of the atmosphere. However, for small scattering angles also the forward scattering properties of aerosols can become important. Thus for small scattering angles (< about 10° to 20°) the uncertainties caused by the incorrect description of the phase function can also become important, and the results for such viewing geometries should be treated with caution. Additional investigations are needed to quantify the respective uncertainties for small scattering angles. The discussion is included in the revised version of the manuscript (P14, L332-334).

5. If I understand the authors correctly, the instrument measures in an altitude of 80m. How is this "elevated" position treated by the algorithms? What is the meaning of the lowermost retrieval grid point in this context?

The retrieved profiles for both algorithms are provided for the altitude range 0 – 4 km with 200 m vertical resolution refer to altitudes above the instrument. The term 'surface concentration' does not refer to the concentration directly at ground level, but to the mean near-surface concentrations in the first 200 m above the instrument. This is done because the profile parameterization used within MAPA allows for the

retrieval of lifted trace gas layers for a shape parameter (s) greater than 1. In such cases, the surface concentration of the parameterized profile would be zero and the comparison with the in-situ measurements would lead to low-biased MAPA results. Therefore there was no need for extrapolation to obtain the 'surface concentrations'. This approach is also used in other studies (e.g., Tirpitz et al., 2021). The altitude of all the remote sensing instruments is 60 m (and not 80 m, as was wrongly stated) and is given relative to the sea level. The in situ measurements of $NO_2$ are performed at a site that is located 174 m above sea level, i.e., 114 m above the MAX-DOAS instrument, and is well within the first 200 m layer retrieved by the MAX-DOAS.

Specific comments

P2, L43: "can lead to or" ) can lead to ... or deteriorate ...

Done.

P3, L81 - L84: You mention that the data is also analyzed regularly within the FRM4DOAS project. Please name the specific differences in retrieval settings between your study and the regularly submitted data and the reason for specific changes of settings. It would also be interesting to compare FRM4DOAS data with your new settings.

Done: A discussion is included in the paper.

Fig. 2: Please add other instruments if not measured at the same location (e.g. in situ).

Done: In-situ measurement site location is now also included.

P10, L211: "by assuming a correlation length" ) "by assuming a Gaussian function with correlation length of...". Note that a correlation length of 50m was used in the cited publication!

Done.

P10, L224: What is the lowermost retrieval altitude for each algorithm? Surface values were extrapolated?

Please see our response for the 5[th] point above.

P12, L285: Why did you use hourly mean values? You could also average all in situ values for the corresponding MAX-DOAS elevation scan cycles.

The suggestion made by the reviewer is very reasonable. However, this would not be possible in our case because hourly mean $NO_2$ concentration is the highest temporal resolution that is available for the in-situ measurements from the air-quality network stations.

P13, L288 - L292: I don't understand your reasoning here. I guess that tracffic emissions contribute strongly to the MAX-DOAS signal but then an in situ site should not be a background site. How far away is the next site in viewing direction of the telescope?

The municipal network of air-quality stations that are distributed around the city center are installed very close to the ground (sampling inlet at ~3 m) and are strongly affected by local traffic emissions. Furthermore, they are located closer to the sea level and are therefore about 30 – 40 m below the MAX-DOAS. Due to its higher altitude, the MAX-DOAS system measurements are considered more representative of the $NO_2$ concentrations in the local boundary layer and less influenced by the local traffic of the city center. The selected for the comparison air-quality site, which is located 114 m above the MAX-DOAS system, probes air that is also more representative of the local boundary layer.

P13, L304: 5° is already quite small, especially when using Henyey-Greenstein. Have you tried different values? I would expect that 10° improves data quality significantly but might decrease the number of data points (maybe too much?).

At this point, we wanted to exclude from the analysis the elevation scans that have been performed very close to the solar azimuth angle due to potential errors of RTM simulations. The limit of 5° was selected arbitrarily in order to keep a balance: The elevation scans close to the solar position are rejected and in the meantime there is no major loss of the data. As Reviewer 1 already mentions, using $10^o$ or more as relative azimuth angle filtering would lead to a significant reduction of the data points number.

P13, L308-L309: I am wondering how negative columns can make it through any flagging step? Also 8.5% is a really large fraction of invalid profiles. Is there any reason known why MAPA produces so many unrealistic profiles?

Due to the low SNR in the UV, larger scatter of the retrieved dSCDs (especially for HCHO) is expected and consequently also of the corresponding profiles. Retrievals of negative columns are not necessarily flagged as invalid by default in order to keep the means unbiased. By doing so, in cases where no tropospheric trace gas is present the retrieved profiles would then contain pure noise. Otherwise, the resulting mean would be high-biased (greater than zero).

P14, Table 3: When looking at the HCHO fraction (also aerosols in UV) of valid profiles for MAPA, I am really worried about the general performance of MAPA in the UV. Is there any particular reason for this bad performance? There was already a BIAS found for MAPA's HCHO results in Tirpitz et al. 2021 so I don't think that noisy data can explain everything!

Since in MAPA retrievals no a priori constraints are used, more strict flagging needs to be applied for retrieved dSCDs that are characterized by large uncertainties (e.g., due to larger fit error or the effects of clouds). As a result, a smaller fraction of the data is flagged as valid. Especially for HCHO, the apparent worse performance of MAPA could be explained by the lower SNR in the UV, along with the higher HCHO profile height compared to $NO_2$ and the decreasing sensitivity towards higher altitudes. Moreover, the trace gas retrievals depend also on the aerosol retrievals, so the respective uncertainties for high AODs would be larger than for low AOD.

P15, Figure 5: For reach row, MAPA shows values close to zero, except for NO2. I am not sure if it is a good thing, that MMF doesn't show small values at all or that MAPA cannot find them only for NO2. Could you please say something about that? And again, I would be interested in MAPAS flagging thresholds and if they differ in the visible and UV spectral range.

The sensitivity of the MAX-DOAS decreases with altitude and it is very limited at altitudes above 2.5 km for the species measured in the VIS spectral range or even lower (1.5 km) for the species in the UV (Figure 6 [8]). For $NO_2$, this is generally not a problem since the total column is dominated by the concentration in the lower layers of the troposphere (see also Figure 9 and discussion 3.5 of the revised version of the manuscript). However, HCHO can be vertically extended at higher altitudes, where the sensitivity of the MAX-DOAS is low. In the case of HCHO, MMF is more prone to result in the a priori profile, while MAPA retrievals become more unstable. Thus, the vertical profiles of MAPA are expected to have greater variability. For aerosols, the discrepancies between MMF and MAPA retrievals are also due to an additional factor: The variable $O_4$ scaling factor that is included in MAPA ($O_4$ SF = var), while no scaling factor is applied to MMF retrievals ($O_4$ SF = 1).

P15, Figure 6: I think this figure tells us that MMF has a positiv Bias for low elevation angles (reddish dots more often over black line) which would also explain why we don't see small values in Fig. 5 for MMF. It seems that the algorithm has problems in retrieving accurate profiles for small dSCD, especially in the UV. This could be explained by more noise but the MAPA results seem to be unaffected. Do you have any explanation for the different LOS depending performance of both algorithms?

We assume that this comment of the reviewer is associated with the $O_4$ dSCDs (especially in the UV) rather than the trace gas dSCDs. The positive bias for low elevation angle that the reviewer mentions is not apparent for $NO_2$ and neither for HCHO, where the data is much noisier. It is true that MAPA results seem to be less affected. The main driver for this behavior would be the different $O_4$ scaling factors applied to the retrievals of MMF and MAPA. While MMF does not include a scaling factor, MAPA fits an optimum $O_4$ scaling factor in order to bring measured and simulated dSCDs into better agreement.

P18, L382 - L384: Concentrations for the lowermost layer rather than conc. at ground? Do you mean the lowermost layer with concentrations larger than zero? If not, please explain!

The output grid of both MMF and MAPA ranges from the ground up to 4 km with 200 m vertical resolution. The term "surface concentration" in Sect. 3.3 does not refer to the concentration at ground level and also it does not refer to the lowermost layer with concentrations larger than zero. It refers to the average concentration in the lowest 200 m above the instrument, as retrieved for the MAX-DOAS first profile layer (i.e., 0 – 200 m). The text has been revised.

P18, Figure 7: Again, MMF doesn't show HCHO values close to zero which means that the main HCHO concentration is found in higher altitudes. MAPA seems to retrieve HCHO closer to the surface. However, P18, L382 - L384 tells us that this conclusion might be wrong. So I am wondering if you could show a similar figure with surface concentrations only? I have to admit that I am confused by the sentence P18, L382 - L384 and the fact that MAPA finds HCHO concentrations close to zero!

The non-zero near-surface MMF HCHO results are probably a result of the non-zero surface concentrations in the a priori profile. They are not a direct consequence of possibly enhanced HCHO concentrations at higher altitudes (note that both retrieval results use the same input measurements; thus the information content is the same for MAPA and MMF).

Concerning the statement in [P18, L382 - L384]: It is true that the surface values are also influenced by HCHO concentrations at higher altitudes. But for the retrieval results close to the surface, the sensitivity of the MAX-DOAS measurements clearly peaks for HCHO close to the surface. Thus the influence of HCHO at high altitudes (> 500m) is usually rather small.

P21, L439 - L440: I am not sure if I understand scheme #3 correctly. In this line, you write about warning flags while you use "erroneous" in Table 4. Please describe this scheme more detailed.

Corrected: From "Data that are not flagged erroneous neither by MMF nor by MAPA are considered valid" to "Data that are flagged as warning by either MMF or MAPA are also considered valid". Both MMF and MAPA flag the data as either valid, warning or error. Scheme #3 rejects the error flagged data but treats the warning flagged data as valid.

P24, L493 - L494: "Aerosol layers between 2 and 4 km are "invisible"...". This is not correct! An elevated layer will for sure be identified as elevated layer in these altitude regions if aerosols below are negligible. MAX-DOAS might not find the correct altitude but the elevated layer will be identified for sure showing a small but existing sensitivity.

This statement was used in order to explain the difference in the profiles of the MAX-DOAS and the lidar between 2 and 4 km for this particular case scenario of 05-Jun, but perhaps it could be interpreted as a more generic statement by a reader. We revised this statement to: "Aerosol layers between 2 and 4 km that are detected by the lidar cannot be well retrieved by the MAX-DOAS, due to its limited sensitivity at these altitudes".

Figure 12: It is hard to say which profile is the best, especially for the 21. of July. Could you please add a subplot showing the modelled and measured dSCDs at each elevation angle for all scenarios and both algorithms? Maybe this helps to assess better the performance here.

The figure below shows the measured and simulated $O_4$ dSCDs as a function of the elevation angle (left) and the correlation plots between measured and modeled dSCDs for MMF and MAPA (right). The performance of the forward models is similar with no clear conclusion of which simulates the true state (measured) better. The aerosol extinction profiles that are shown in Figure 13 [12] are retrieved by the MAX-DOAS in the visible range. As seen in Figure 9 (Sect. 3.5 of the revised version) larger discrepancies are found in the profiles of MMF and MAPA during summer. Since these plots are not very helpful in resolving this issue, we have not included them in the revised manuscript.

[Figure]

Figure A3: As you have mentioned, the error bars for the scaling factors are larger in winter than in summer.I was wondering if the number of data points in winter is large enough to show a mean daily variation for January (and compare with a similar curve from August)? Do these curves show a clear diurnal cycle?

In this study we do not investigate the diurnal cycle of the $O_4$ scaling factor, because no clear conclusion can be drawn, mainly due to insufficient number of data during winter, as the reviewer has pointed out. This study is based on approximately 1 year of data. It should be noted also that only the $O_4$ SFs for which $65^o < SZA < 75^o$ are presented. Nevertheless, despite of the large error bars, an apparent seasonal pattern is observed that could be further investigated when longer time series of MAX-DOAS measurements become available.

**References**

Beirle, S., Dörner, S., Donner, S., Remmers, J., Wang, Y., and Wagner, T.: The Mainz profile algorithm (MAPA), Atmospheric Measurement Techniques, 12, 1785–1806, https://doi.org/10.5194/amt-12-1785-2019, 2019.

Tirpitz, J.-L., Frieß, U., Hendrick, F., Alberti, C., Allaart, M., Apituley, A., Bais, A., Beirle, S., Berkhout, S., Bognar, K., Bösch, T., Bruchkouski, I., Cede, A., Chan, K. L., den Hoed, M., Donner, S., Drosoglou, T., Fayt, C., Friedrich, M. M., Frumau, A., Gast, L., Gielen, C., Gomez-Martín, L., Hao, N., Hensen, A., Henzing, B., Hermans, C., Jin, J., Kreher, K., Kuhn, J., Lampel, J., Li, A., Liu, C., Liu, H., Ma, 820 J., Merlaud, A., Peters, E., Pinardi, G., Piters, A., Platt, U., Puentedura, O., Richter, A., Schmitt, S., Spinei, E., Stein Zweers, D., Strong, K., Swart, D., Tack, F., Tiefengraber, M., van der Hoff, R., van Roozendael, M., Vlemmix, T., Vonk, J., Wagner, T., Wang, Y., Wang, Z., Wenig, M., Wiegner, M., Wittrock, F., Xie, P., Xing, C., Xu, J., Yela, M., Zhang, C., and Zhao, X.: Intercomparison of MAX-DOAS vertical profile retrieval algorithms: studies on field data from the CINDI-2 campaign, Atmospheric Measurement Techniques, 14, 1–35, https://doi.org/10.5194/amt-14-1-2021, 2021.

---

## Author Comment (AC2)

Response to Referee #2

We would like to thank the reviewer for carefully reading the manuscript and for providing helpful comments, remarks and suggestions. You can find below our responses in red after each individual comment:

Due to the reordering of Sect. 3 (Results) and adding new figures, all references to text and figures of the manuscript that are mentioned below refer to the revised version, while in square brackets [] they refer to the preprint.

e.g., Figure 6 [8] refers to Figure 6 of the revised version and to Figure 8 of the preprint.

The paper "Retrieval of tropospheric aerosol, NO2 and HCHO vertical profiles from MAX-DOAS observations over Thessaloniki, Greece" by Dimitris Karagkiozidis et al., presents results of 2 MAXDOAS profiling retrievals (MMF and MAPA) for 1 year of observation (May 2020 to May 2021) in Thessaloniki. The 2 approaches are presented, with investigations of the impact of different filtering selections, and are compared to available ancillary measurements (AOD from Brewer and CIMEL, aerosols extinction profiles from a few lidar measurements and surface NO2 from in-situ data).

The paper is well written and easy to follow, and its scientific content fits the scope of AMT.

The title is however a bit misleading: we expect to learn about profiles in Thessaloniki, but NO2 and HCHO profiles are never shown in any of the figures! The paper is more about a comparison of the 2 approaches, mostly focusing on VCD and surface concentration, and comparisons to external data, when available (which is not the case for HCHO). The outcome of the study is also a bit confusing, specifying for each case the best regression statistics, but not how to deal with these data if they want to be used. Should an average of both profiling techniques should be recommended? Should we only rely on VCD and surface concentration? Should we use only one of them (eg MMF that provides AVK), but then use the bias to MAPA to estimate a (more) realistic uncertainty? Are the profiles of the 2 algorithm within their estimated uncertainties? (uncertainties of each approach are never mentioned).

It would be good that the authors provide some suggestions in the conclusions.

I would thus recommend some revision of the title and text, with some further geophysical (instead of only statistical) investigation, as described below.

I would also suggest some reordering of Section 3. The results are now presented first for VCD (3.1), then dSCD (3.2), then surface concentration (3.3) and then AVK (3.4). It would make more sense to me to follow the retrieval order: from dSCD, to profiles and AVK, and then VCD and surface extracted from the profiles. Or focusing first on the output products (VCD and surface concentration), and then some diagnostic elements (dSCD and AVK).
* * *
The study would allow to present many geophysically results, instead of only showing coherence of 2 (both are possible, see eg. Vlemmix et al., 2015). E.g. answering the following questions:

- how are the profiles themselves (is e.g. the H75 characteristic height (see eg Vlemmix et al., 2015) of NO2 lower or higher than the HCHO and aerosols one? how is it changing over the day and the seasons?)

- how is the variability within the different measured azimuths (is there an heterogeneous situation, as shown e.g. for Athens in Gratsea et al 2016? is it stronger for NO2 than for HCHO, as we would expect?).

The general comments made by Referee #2 (e.g., misleading title, vertical profiles not shown, geophysical results not presented) are very reasonable and are similar to those made by Referee #1. Based on Referee's #1 suggestions, we considered more appropriate that the paper remains in an intercomparison/validation scope, yet including additional information. The title of the paper was revised to: "Retrieval of tropospheric aerosol, NO$_2$ and HCHO vertical profiles from MAX-DOAS observations over Thessaloniki, Greece: Intercomparison and validation of two inversion algorithms". We included a subsection in Sect. 3 (Results) showing the seasonal mean vertical profiles retrieved by MMF and MAPA for all species. In this subsection the vertical profiles are intercompared, the variability of each algorithm is discussed along with possible sources for each species. Section 3 was reordered according to Referee's #2 suggestions: (1) dSCDs, (2) Averaging kernels, (3) VCDs, (4) Surface concentrations, (5) Seasonal mean vertical profiles.

Also, to my point of view, the paper is missing the opportunity to make the link with the previously created datasets from this instrument. It would be nice to know how much these profiling results are coherent with approaches used in the past for the VCD estimation (Drosoglou et al., 2017 and 2018, QA4ECV dataset used in Pinardi et al., 2020; Verhoelst et al, 2021; De Smedt et al. 2021). Are results similar or very different in term of VCD? E.g., see comment for P 14, line 338, or for P. 20, line 412.

The instrument that is used in this study was installed and its operation began in May 2020. Data from this specific instrument have not been used in former studies and have not been submitted to any databases (except for the FRM$_4$DOAS). A direct comparison with data that have been used in the past (e.g., Pinardi et al., 2020; Verhoelst et al, 2021; De Smedt et al. 2021) is not possible since these data have been retrieved using instruments of different characteristics (e.g., tracker resolution, wavelength range, spectral resolution, SNR, FOV). However, we included an extra appendix showing the comparison of the NO$_2$ and HCHO VCDs that are retrieved by MMF and MAPA with the VCDs that are calculated using the geometric approximation. The VCDs that have been used in Drosoglou et al., 2017 and 2018 were retrieved using pre-calculated dAMF LUTs based on RTM simulations. The NO$_2$ dAMFs are calculated at a wavelength that corresponds to the smaller fitting window of NO$_2$ (411-455 nm), because of the limited wavelength range of the older spectrographs. In this study NO$_2$ is retrieved at the large visible range (425-490 nm), so an additional dAMF LUT would be required, which is currently not available.

There is also a lack of reference to literature when presenting the specific results of this study and stating some "realities". (e.g., page 24, line 485 "Since the MAX-DOAS profile retrievals in the UV are sensitive

only at altitudes closer to the ground*, where the lidar system is not, the profiles for 360 nm are excluded from the analysis") - *: how can we confirm this sentence? )

Done: References to literature have been included at the proper sections of the manuscript.

It would be good to also show the coherence of the lidar comparisons (Figure 12) with the AOD from Brewer and AERONET. Is the vertically integrated extinction profile coherent with the AOD? (see comment for Figure 12)

Please see relevant reply below (comment for Figure 13 [12]).
* * *
- Figure 2: please also specify other instruments location.

Done.

- Section 2.3 (or 2.6): are the 2 retrievals treating cloud filtering in the same way? are they both starting from a reduced set of cloud filtered dSCD, or is this done within the MMF and MAPA algorithms?

No cloud filtering is applied to the data prior to the analysis of MMF and MAPA. Neither MMF nor MAPA include a direct cloud flagging system. However, in MMF the stability of the retrieval is internally checked. The aerosols retrieval is performed twice: Once using the a priori profile that is defined in the settings (Sect. 2.6) and once using different a priori (and hence different covariance matrix information). If the retrieved AODs are significantly different, the trace gas retrieval is also performed twice. If the retrieved VCDs are also significantly different the elevation scan is flagged as error. Moderately thin uniform clouds do not prevent good retrievals (Frieß et al., 2019). For very non-uniform cloud conditions, it is very likely, that the two aerosol retrievals will result in very different aerosol profiles and if they have a strong effect on the trace gas retrieval, then the retrieval is flagged as invalid. For MAPA the flagging criteria might be too strict, but it is found in other studies that the elevation sequences affected by clouds are correctly flagged as invalid (Beirle et al., 2019; Frieß et al., 2019; Tirpitz et al., 2021) since some flags that are included in MAPA are sensitive to clouds.

- page 10, line 212: "the progress of the convergence is faster when using an a priori VCD or AOD below the true value" - why is this?

The reason has not been yet identified. This is found empirically, by testing the convergence behavior.

- Section 2.7: specify the location of the ancillary data - how far are they from the MAXDOAS? and mention the impact of the different fields of views.

A short introduction is included now in Sect. 2.7. Except for the in situ, all other instruments that are used in this study (MAX-DOAS, CIMEL, Brewer and lidar) are collocated on the rooftop of the Physics Department of the Aristotle University of Thessaloniki (40.634$^o$ N, 22.956$^o$ E), about 60 m above sea level. The location of the in situ measurement site is at a distance of ~ 1.2 km away from the MAX-DOAS and it now included in Figure 2. Differences in the retrieved products among the instruments are mainly due the viewing geometries and the retrieval technique that each instrument utilizes rather than the field of view (P.12, L251 - 252). The CIMEL and the Brewer use direct sun measurements for the calculation of the AOD, while the MAX-DOAS uses the $O_4$ dSCDs at different elevation angles as a proxy for the retrieval of the aerosol extinction (P.27, L538 - 540 [P22, L450 - L452]). The instruments also probe different air masses, e.g., the lidar measures only at the zenith, while the MAX-DOAS retrieves the vertical profiles by scanning at multiple elevation angles for a certain azimuth direction (P.27, L562 - 565 [P24, L473 - L475]).

- P. 12, line 259: just to have an idea, how many lidar measurements this schedule would represent in the interested time period (May 2020 to May 2021)?

The lidar measurements in Thessaloniki follow mainly the EARLINET schedule for climatological measurements with additional measurements for special events and satellite overpasses, resulting in more than 100 days of data per year. Generally, lidar measurements are only restricted by unfavorable weather conditions (rain, low clouds) and recently by system upgrade. For example, 111 days of measurements are available for the period 2019 - 2020, whilst only 35 measurements were performed between May 2020 to September 2020, when the system was set out of order for upgrading.

- Section 3: the results are presented separately for the different viewing azimuths (with no clear major difference or explation of difference between MMF and MAPA relative to the azimuth), while in Sect. 4, where the results are "validated", this information is now missing. What is used here? only one of the azimuths or an average of both or a mix of them, depending on the time of the day?

Please see the relevant replies on the comments for Figure 13 [12] and Figure 14 [13] below.

- P. 13, line 305: "the elevation sequences, for which the retrieved AOD from the MAX-DOAS inversion algorithms is greater than 1.5 are filtered-out, since such high aerosol loads are unrealistic for Thessaloniki" - is this a big proportion of data? can this be impacted by clouds, or have these been filtered before?

The elevation sequences that are passed to MMF and MAPA have not been filtered for clouds prior to the analysis. Indeed, retrievals of high AOD can be strongly impacted by clouds. The flagging that is applied to the data has proven to successfully reject retrievals under such conditions (e.g., Beirle et al., 2019; Frieß et al., 2019; Tirpitz et al., 2021). Filtering scans that result in AODs greater than 1.5 further assures that unrealistic profiles will be eliminated.

- P. 13, line 307: "Negative columns can occur in the trace gas retrievals of MAPA within the Monte Carlo ensemble and they are intentionally not removed" - add "at first/by default/..." - is this 8.5% of negative HCHO VCD points already included in the 18% valid MAPA flagged data of Table 3, or to be additionally removed ?

The fractions of 8.5% and 18% are not directly comparable. The fraction of 8.5% corresponds to the fraction of the valid-flagged data that contain negative columns, while 18% refers to the fraction of the total dataset that is flagged as valid. Since negative columns are removed from the initial dataset, the fraction of 18% refers to the data that are flagged as valid and do not contain negative concentrations.

- Table 3: add a third column with the remaining valid data percentage when both algorithms have coincident valid flags (filter #4, used as default in most of this section, if I understood well).

Done.

- P. 13, line 317: "an elevation sequence is considered valid as long as it is flagged as valid by both MMF and MAPA. This is the default flagging scheme for NO2, HCHO and AOD at 477 nm" --> this would mean flagging scheme #4 of Table 4, right?

Yes, this is true. Scheme #4 of Table 4 accounts for the data that are flagged as valid by both MMF and MAPA. The text has been revised to clarify this.

- P. 14, line 324: you mention the Orthogonal Distance Regression (ODR or bivariate least-squares) instead of an Ordinary Linear Regression (OLR or standard least-squares), but in figures 5, 7, 11 you mention linear regression. Please adapt with the correct regression type.

Done.

- P 14, line 338 "This is the first time during the Phaethon's operation that the whole elevation sequence is being used in order to derive the tropospheric VCDs more accurately": comment coherence of VCD results obtained here with respect to approaches used in past datasets (see comment above).

An extra appendix in included, showing the comparison of the VCDs that are retrieved by the inversion algorithms (MMF and MAPA) with the VCDs that are obtained using the geometric approximation (a technique that was used in previous datasets. Please see also relevant reply above (2$^{nd}$ half of page 2).

- Figure 5: it is difficult to understand from this figure if the larger variability of MAPA results (eg for HCHO and aerosols UV) is related to the different azimuths - is MMF seeing less well the variability among the different azimuths, is MMF too sensitive or is this a false impression? are the SCD showing some systematic (?) larger signal over the city or the sea? or is this just coming from the larger variability in aerosols in the UV ? (if latter explanation is relevant, also add it to P. 16, lines 355- 356).

In the UV, the MAX-DOAS loses its sensitivity at higher altitudes faster than in the VIS. The sensitivity of the MAX-DOAS decreases with altitude and it is very limited at altitudes above 2.5 km for the species measured in the VIS spectral range or even lower (1.5 km) for the species in the UV (Figure 6 [8]). For $NO_2$, this is generally not a problem since the total column is dominated by the concentration in the lower layers of the troposphere (see also Figure 9 and discussion 3.5 of the revised version of the manuscript). However, HCHO can be vertically extended at higher altitudes, where the sensitivity of the MAX-DOAS is low. In the case of HCHO, OEM algorithms (such as MMF) are more prone to result in the a priori profile, while parameterized algorithms (such as MAPA) become more unstable (Frieß et al., 2019). Thus, the vertical profiles of MAPA are expected to have greater variability.

- P. 19, line 396: consider "Figure 8 shows a typical example of the calculated AVKs for each of the retrieved species. The DoF of this example retrieval are shown for each species." --> "Figure 8 shows a typical example of the calculated AVKs for each of the retrieved species, including their corresponding DoF."

Done.

- P. 19, line 399: "The averaging kernels verify that" - change "verify" to "illustrate" or something similar.

Done

- P. 20, line 412: no other source of independent HCHO is present, but this section could also be a good place to compare results from the 2 profiling algorithms to results of past VCD retrieval methods (see comment above)

Please see relevant reply above (2nd half of page 2).

- Figure 9: what flagging choice is used to make this figure? from this figure, the feeling is that MAPA has systematically lower AOD @477 than the other datasets (a lot of points close to zero), which is not the case for AOD @360. I would say that the comparisons in the UV are better than in the visible...

We would like to thank the reviewer for pointing this, because there is an inconsistency in the time series of aerosols (UV) between Figure 10 [9] and Figure 7 [5]. The time series in Figure 10 [9] were supposed to use data flagged with the default flagging schemes, as described in Sect. 3 (i.e., scheme #2 of Table 4

for aerosols UV and #4 for aerosols VIS), yet, accidentally, scheme #4 was used for both species. The time series of aerosols UV is now corrected and is consistent with the rest of the manuscript. Figure 10 [9] is included to depict the different time periods that the three systems (MAX-DOAS, CIMEL and Brewer) cover during this ~1 year study. A more detailed comparison (not just visual) of the AODs between the MAX-DOAS and CIMEL/Brewer, including all flagging schemes of Table 4, is presented in Sect. 4.1 and Sect. 4.2.

- P. 21, line 435: "Compared to the CIMEL, MAPA seems to perform slightly better than MMF when its own flagging algorithm is applied to the data, with correlation coefficients of 0.70 and 0.50, respectively." - suggestion to replace by "when each algorithm consider is own flagging, with correlation coefficients of 0.70 and 0.50, respectively (MAPA for case #2 and MMF for case #1)." for more clarity!

Done.

- P. 21, line 447: "The AOD derived from the MAX-DOAS, both in the UV and the VIS range, is, generally, underestimated compared to the AOD measured by the CIMEL and the Brewer" --> add references to other studies showing that! also in P. 22, line 454.

Done.

- P. 24, line 476: "Thus, differences in the retrieved extinction profiles are expected, especially at locations with large horizontal inhomogeneities of aerosols" --> is this the case here? having a geophysically analysis (diurnal and seasonal) of the results for the different azimuths would help answer to this question. What azimuth is shown in Figure 12 for MAXDOAS?

The MAX-DOAS aerosol extinction profiles in Figure 13 [12] correspond to the elevation scans which are flagged as valid by both MMF and MAPA (scheme #4 of Table 4) that are closest in time to the lidar measurements. For the cases of 04 and 05 June 2020, the MAX-DOAS system was not scanning across all viewing directions (P.18, L418 - 420 [P. 16, L347 – 349]), so only 220$^o$ azimuth is available. For the other two cases (i.e., 21-Jul-2020 and 28-Aug-2020), the elevation scans correspond to 185 and 220$^o$ azimuths, respectively. The azimuth viewing direction of the MAX-DOAS are now included in the title of each panel.

- Figure 12: it would be nice to also have a comparison of the integrated aerosols profiles, to compare the lidar AOD to the MAXDOAS ones and to Brewer and AERONET (if available) for those cases.

AOD measurements from the CIMEL and/or the Brewer are not available for all cases shown in Figure 13 [12] for reasons that are discussed in Sect. 4 (see also Figure 10 [9]). However, the consistency of lidar and CIMEL AOD measurements over Thessaloniki was analyzed in the study of Siomos et al., 2018 using fourteen years of data. Periodical systematic biases (e.g., lidar overlap effect) that could affect the

annual cycles, non-periodical biases that could interfere with the long-term trends and possible effects of the different sampling rate between the lidar and the sunphotometer were discussed and analyzed. The analysis resulted in consistent statistically significant and decreasing trends of aerosol optical depth (AOD) at 355 nm of −23.2 and −22.3 % per decade for the lidar (integrated extinction coefficients) and the sunphotometer datasets, respectively (Siomos et al., 2018). The AODs at 355nm measured by the lidar have also been compared with the Brewer's retrievals, showing a generally good correlation of 0.7 (Voudouri et al., 2017).

- P. 26, line 510: what MAXDOAS dataset is shown in Figure 13? all the azimuth angles together?

Since the in situ site is not located in the MAX-DOAS line of sight, hourly mean $NO_2$ surface concentrations from all available azimuth directions are calculated in order to avoid effects of possible horizontal inhomogeneities of $NO_2$.

- Figure 13: how is the fact that the MAXDOAS is situated at an height of 80m is taken into account here?

The MAX-DOAS system is located at an altitude of 60 m above sea level (not 80 m as was wrongly stated), while the in situ monitoring station is located at 174 m above sea level. The $NO_2$ "surface concentrations" reported by the MAX-DOAS refer to the average concentration retrieved for the lowermost 200 m layer above the MAX-DOAS location, so the in situ sampling is well within the first MAX-DOAS layer.

- P. 26, line 517: Zieger et al 2011 reference is for aerosols comparisons, it should appear in Sect. 4.1 instead of 4.2

Done.

- Figure A1: why none of the statistics for NO2 and HCHO correspond to those of Figure 7 black values? I would assume to find the same values in "O4 SF var"?!

The reviewer correctly realized that the statistics of the $O_4$ SF var in Figure A1 should match with the corresponding statistics of the VCDs/AODs (when no discrimination of the different azimuth viewing directions is made) that were presented in Sect. 3 (Results). However, Figure A1 presents the effect of the $O_4$ SF on the integrated columns (i.e., VCDs and AODs), not on the surface concentrations. Hence, the statistics of the $O_4$ SF var in Figure A1 should match with those of Figure 7 [5] (black values) and not with Figure 8 [7].

**References**

Beirle, S., Dörner, S., Donner, S., Remmers, J., Wang, Y., and Wagner, T.: The Mainz profile algorithm (MAPA), Atmospheric Measurement Techniques, 12, 1785–1806, https://doi.org/10.5194/amt-12-1785-2019, 2019.

De Smedt, I., Pinardi, G., Vigouroux, C., Compernolle, S., Bais, A., Benavent, N., Boersma, F., Chan, K.-L., Donner, S., Eichmann, K.-U., Hedelt, P., Hendrick, F., Irie, H., Kumar, V., Lambert, J.-C., Langerock, B., Lerot, C., Liu, C., Loyola, D., Piters, A., Richter, A., Rivera Cárdenas, C., Romahn, F., Ryan, R. G., Sinha, V., Theys, N., Vlietinck, J., Wagner, T., Wang, T., Yu, H., and Van Roozendael, M.: Comparative assessment of TROPOMI and OMI formaldehyde observations and validation against MAX-DOAS network column measurements, Atmos. Chem. Phys., 21, 12561–12593, https://doi.org/10.5194/acp-21-12561-2021, 2021.

Drosoglou, T., Bais, A. F., Zyrichidou, I., Kouremeti, N., Poupkou, A., Liora, N., Giannaros, C., Koukouli, M. E., Balis, D., and Melas, D.: Comparisons of ground-based tropospheric NO2 MAX-DOAS measurements to satellite observations with the aid of an air quality model over the Thessaloniki area, Greece, Atmos. Chem. Phys., 17, 5829–5849, https://doi.org/10.5194/acp-17-5829-2017, 2017.

Drosoglou, T., Koukouli, M. E., Kouremeti, N., Bais, A. F., Zyrichidou, I., Balis, D., van der A, R. J., Xu, J., and Li, A.: MAX-DOAS NO2 observations over Guangzhou, China; ground-based and satellite comparisons, Atmos. Meas. Tech., 11, 2239–2255, https://doi.org/10.5194/amt-11-2239-2018, 2018.

Frieß, U., Beirle, S., Alvarado Bonilla, L., Bösch, T., Friedrich, M. M., Hendrick, F., Piters, A., Richter, A., van Roozendael, M., Rozanov, V. V., Spinei, E., Tirpitz, J.-L., Vlemmix, T., Wagner, T., and Wang, Y.: Intercomparison of MAX-DOAS vertical profile retrieval algorithms: studies using synthetic data, Atmos. Meas. Tech., 12, 2155–2181, https://doi.org/10.5194/amt-12-2155-2019, 2019.

Pinardi, G., Van Roozendael, M., Hendrick, F., Theys, N., Abuhassan, N., Bais, A., Boersma, F., Cede, A., Chong, J., Donner, S., Drosoglou, T., Dzhola, A., Eskes, H., Frieß, U., Granville, J., Herman, J. R., Holla, R., Hovila, J., Irie, H., Kanaya, Y., Karagkiozidis, D., Kouremeti, N., Lambert, J.-C., Ma, J., Peters, E., Piters, A., Postylyakov, O., Richter, A., Remmers, J., Takashima, H., Tiefengraber, M., Valks, P., Vlemmix, T., Wagner, T., and Wittrock, F.: Validation of tropospheric NO2 column measurements of GOME-2A and OMI using MAX-DOAS and direct sun network observations, Atmos. Meas. Tech., 13, 6141–6174, https://doi.org/10.5194/amt-13-6141-2020, 2020.

Siomos, N., Balis, D. S., Voudouri, K. A., Giannakaki, E., Filioglou, M., Amiridis, V., Papayannis, A., and Fragkos, K.: Are EARLINET and AERONET climatologies consistent? The case of Thessaloniki, Greece, Atmos. Chem. Phys., 18, 11885–11903, https://doi.org/10.5194/acp-18-11885-2018, 2018.

Tirpitz, J.-L., Frieß, U., Hendrick, F., Alberti, C., Allaart, M., Apituley, A., Bais, A., Beirle, S., Berkhout, S., Bognar, K., Bösch, T., Bruchkouski, I., Cede, A., Chan, K. L., den Hoed, M., Donner, S., Drosoglou, T., Fayt, C., Friedrich, M. M., Frumau, A., Gast, L., Gielen, C., Gomez-Martín, L., Hao, N., Hensen, A., Henzing, B., Hermans, C., Jin, J., Kreher, K., Kuhn, J., Lampel, J., Li, A., Liu, C., Liu, H., Ma, 820 J.,

Merlaud, A., Peters, E., Pinardi, G., Piters, A., Platt, U., Puentedura, O., Richter, A., Schmitt, S., Spinei, E., Stein Zweers, D., Strong, K., Swart, D., Tack, F., Tiefengraber, M., van der Hoff, R., van Roozendael, M., Vlemmix, T., Vonk, J., Wagner, T., Wang, Y., Wang, Z., Wenig, M., Wiegner, M., Wittrock, F., Xie, P., Xing, C., Xu, J., Yela, M., Zhang, C., and Zhao, X.: Intercomparison of MAX-DOAS vertical profile retrieval algorithms: studies on field data from the CINDI-2 campaign, Atmospheric Measurement Techniques, 14, 1–35, https://doi.org/10.5194/amt-14-1-2021, 2021.

Verhoelst, T., Compernolle, S., Pinardi, G., Lambert, J.-C., Eskes, H. J., Eichmann, K.-U., Fjæraa, A. M., Granville, J., Niemeijer, S., Cede, A., Tiefengraber, M., Hendrick, F., Pazmiño, A., Bais, A., Bazureau, A., Boersma, K. F., Bognar, K., Dehn, A., Donner, S., Elokhov, A., Gebetsberger, M., Goutail, F., Grutter de la Mora, M., Gruzdev, A., Gratsea, M., Hansen, G. H., Irie, H., Jepsen, N., Kanaya, Y., Karagkiozidis, D., Kivi, R., Kreher, K., Levelt, P. F., Liu, C., Müller, M., Navarro Comas, M., Piters, A. J. M., Pommereau, J.-P., Portafaix, T., Prados-Roman, C., Puentedura, O., Querel, R., Remmers, J., Richter, A., Rimmer, J., Rivera Cárdenas, C., Saavedra de Miguel, L., Sinyakov, V. P., Stremme, W., Strong, K., Van Roozendael, M., Veefkind, J. P., Wagner, T., Wittrock, F., Yela González, M., and Zehner, C.: Ground-based validation of the Copernicus Sentinel-5P TROPOMI NO2 measurements with the NDACC ZSL-DOAS, MAX-DOAS and Pandonia global networks, Atmos. Meas. Tech., 14, 481–510, https://doi.org/10.5194/amt-14-481-2021, 2021.

Voudouri, K.A; Siomos, N.; Giannakaki, E.; Amiridis, V.; D'Amico, G.; Balis, D. Long-Term Comparison of Lidar Derived Aerosol Optical Depth Between Two Operational Algorithms and Sun Photometer Measurements for Thessaloniki, Greece. In Perspectives on Atmospheric Sciences; Springer: Cham, Swizerland, 2017, https://link.springer.com/chapter/10.1007%2F978-3-319-35095-0_113.

---

## Author Response (AR2)

**Response to the Associate Editor**

We would like to thank the Associate Editor for carefully reading the manuscript and for providing helpful comments, remarks and suggestions. You can find below our response.

**The comments from the two reviewers have been addressed satisfactorily in the responses. However, it is not always clear what has been changed in the revised manuscript in reaction to the reviewers' comment. For example, the question of Rev. #2 about the cloud filtering gets an extensive answer but it is not said what has been changed in the manuscript to clarify the issue. Please check the responses, and briefly indicate what action has been taken to address the reviewers' comments.**

Done: In response to the question of Reviewer #2 about the cloud filtering a short discussion is included in P.15, L352-354. Also in response to the question of Reviewer #1 about the in situ measurement site a short discussion is included in P.14, L314-315 of the revised manuscript.

**- Typography in Equation 2: symbol A should be in italics. However, if A is a matrix, it should be bold and upright. If x is a vector, it should be bold and upright.**

The symbols of Equation 2 have been corrected.

**- References: please correct the alphabetical order in some places: Friedrich before Friess, and other places.**

Done: Friedrich is listed before Friess and Holben before Hoenninger in the revised version of the manuscript. This was due to a technical issue. Please note that these corrections do not show up in the track-changes file, because the reference layout is automatically produced after building the source .tex file and therefore latexdiff cannot find the differences.